# An assessment of two automated snow water equivalent instruments during the WMO Solid Precipitation Intercomparison Experiment

Craig D. Smith[1], Anna Kontu[2], Richard Laffin[3], John W. Pomeroy[4]

[1]Environment Canada, Saskatoon, S7N 3H5, Canada

[2]Finnish Meteorological Institute, Sodankylä, FI-99600, Finland

[3]Campbell Scientific, Edmonton, T5L 4X4, Canada

[4]Centre for Hydrology, University of Saskatchewan, Saskatoon, S7N 5C8, Canada

Correspondence to: Craig D. Smith (craig.smith2@canada.ca)

## Abstract

During the World Meteorological Organization (WMO) Solid Precipitation Intercomparison Experiment (SPICE),

automated measurements of snow water equivalent (SWE) were made at the Sodankylä (Finland),

Weissfluhjoch (Switzerland) and Caribou Creek (Canada) SPICE sites during the Northern Hemisphere winters of

2013/2014 and 2014/2015. Supplementary intercomparison measurements were made at Fortress Mountain

(Kananaskis, Canada) during the 2013/2014 winter. The objectives of this analysis are to compare automated

SWE measurements with a reference, comment on their performance, and where possible to make

recommendations on how to best use the instruments and interpret their measurements. Sodankylä, Caribou

Creek and Fortress Mountain hosted a Campbell Scientific CS725 passive gamma radiation SWE sensor.

Sodankylä and Weissfluhjoch hosted a Sommer Messtechnik SSG1000 snow scale. The CS725 operating

principle is based on measuring the attenuation of soil emitted gamma radiation by the snowpack and relating

the attenuation to SWE. The SSG1000 measures the mass of the overlying snowpack directly by using a
weighing platform and load cell. Manual SWE measurements were obtained at the intercomparison sites on a
bi-weekly basis over the accumulation/ablation periods using bulk density samplers. These manual
measurements are considered to be the reference for the intercomparison. Results from Sodankylä and
Caribou Creek showed that the CS725 generally overestimates SWE as compared to manual measurements by
roughly 30 to 35 % with correlations ($r^2$) as high as 0.99 for Sodankylä and 0.90 for Caribou Creek. The RMSE
varied from 30 to 43 mm water equivalent (mm w.e.) and 18 to 25 mm w.e. at Sodankylä and Caribou Creek,
which had respective SWE maximums of approximately 200 mm w.e. and 120 mm w.e. The correlation at
Fortress Mountain was 0.94 (RMSE of 48 mm w.e. with a maximum SWE of approximately 650 mm w.e.) with
no systematic overestimation. The SSG1000 snow scale, having a different measurement principle, agreed
quite closely with the manual measurements at Sodankylä and Weissfluhjoch throughout the periods when
data were available ($r^2$ as high as 0.99 and RMSE from 8 to 24 mm w.e. at Sodankylä and 56 to 59 mm w.e. at
Weissfluhjoch, where maximum SWE was approximately 850 mm w.e.). When the SSG1000 was compared to
the CS725 at Sodankylä, the agreement was linear until the start of ablation period when the positive bias in
the CS725 increases substantially relative to the SSG1000. Since both Caribou Creek and Sodankylä have sandy
soil, water from the snowpack readily infiltrates into the soil during melt, even if the soil is frozen.  However,
the CS725 does not differentiate this water from the un-melted snow. This issue can be identified, at least
during the late spring ablation period, with soil moisture and temperature observations like those measured at
Caribou Creek. With a less permeable soil and surface runoff, the increase in the instrument bias during
ablation period is not as significant, as shown by the Fortress Mountain intercomparison.

# 1 Introduction

The measurement of snow water equivalent (SWE) is vital for flood and water resource forecasting, drought monitoring, climate trend analysis, and hydrological and climate model initialization (Barnett et al., 2005; Gray et al., 2001; Bartlett et al., 2006; Laukkanen, 2004). Many of these applications require accurate and timely information about how much water is being held within the snowpack (Pomeroy and Gray, 1995). SWE measurements can be made in-situ, either manually or via automated instrumentation, or derived from remote sensing platforms, and are usually expressed as units of mass per area (kg m$^{-2}$) or in equivalent units of millimetres of water equivalent (mm w.e.). Manual measurements of SWE are typically made using a multi-point bulk density sampling technique along an established transect or snow course (WMO, 2008). Snow course measurements are often time consuming and expensive, especially if required in remote locations (Pomeroy and Gray, 1995). This means that manual SWE measurements may be infrequent or only undertaken when the snowpack is estimated to be at its seasonal maximum. Prohibitive costs of manual snow course observations have led to the reduction of these measurements by many agencies, including Environment and Climate Change Canada, where operational snow course numbers have decreased from over 100 in the 1980s to less than 30 (Barry, 1995; Brown et al, 2000). Since the early 1990s, manual SWE measurements have been augmented or replaced by remote sensing techniques such as passive microwave retrievals (Goodison and Walker, 1995) but these techniques still require accurate and reliable in-situ measurements for ground-truthing and retrieval development (Derksen et al., 2005; Takala et al., 2011).

With the reduced availability of manual SWE measurements, automated instruments for the measurement of SWE are becoming more necessary and more commonplace. Snow pillows have been used for the automated measurement of SWE in remote locations since the 1960s (Beaumont, 1965) by measuring the overlying pressure of the snowpack on a fluid filled bladder. The SNOTEL network in the United States is based on snow pillow measurements (Serreze et al., 1999). More recently, similar measurements are obtained using snow

scales that use a weighing surface and load cell to measure the weight of the overlying snow (Beaumont, 1966;
Johnson et al., 2007; Johnson et al., 2011)). Several indirect methods exist to measure SWE that include the use
of neutron probes (Harding, 1989) in which a radiation source is placed under the snowpack and the scattering
of neutrons through the snow is measured by a detector. Cosmic ray proton probes (Kodama et al., 1979;
Rasmussen et al., 2012) work in a similar manner but do not require an active source. The probes described by
Kodama et al. are installed under the snow while the system described by Rasmussen et al. (called COSMOS) is
installed above the snow. Kinar and Pomeroy (2007; 2015a) outline a method of non-invasive sonic
reflectometry through the snowpack to determine snow density, liquid water content, and temperature. Other
passive radiation sensors are mounted above the surface and measure the attenuation of naturally emitted
radiation from the soil as it passes through the snowpack and then relate this attenuation to SWE content
(Choquette et al., 2008; Martin et al., 2008). Each of these instruments and techniques have advantages and
disadvantages, which are not discussed here (see Kinar and Pomeroy (2015b) for a more comprehensive
description of snow measurement methods and related issues). Rather, this analysis assesses the use and
accuracy of two instruments that were tested during the World Meteorological Organization (WMO-Solid
Precipitation Intercomparison Experiment (SPICE) (Nitu et al., 2012; Rasmussen et al., 2012), namely the
Campbell Scientific CS725 and the Sommer Messtechnik SSG1000 snow scale.
The CS725 (previously known as GMON or GMON3) has been previously field tested by Hydro Québec
(Choquette et al., 2008; Martin et al., 2008) as well as by Wright et al. (2011). Results by Choquette et al. (2008)
showed an average error of +18 % when comparing to 8 manual snow cores over 3 seasons in Quebec. They
obtained a somewhat better agreement with total SWE calculated from density profiles (with an average error
of +5 %) but only had 4 samples over 2 seasons. Wright et al. (2011) showed intercomparison results between
GMON3 sensors and snow pillows, precipitation gauges, and snow courses at Sunshine Village (Alberta,
Canada) and Tony Grove Ranger Station (Utah, USA). Results showed high correlations between the sensor and
(unadjusted) accumulated precipitation (0.99) and between the sensor and snow pillow observations (0.99) but
lower correlations (0.83) with snow course observations (during one season at Sunshine Village). The authors
question the quality and inherent biases in the snow course samples but do not comment on the sources of
error or the proximity of the snow course to the instrument.
Instrument intercomparisons that included the SSG1000 have been limited but some results are reported by
Stranden and Grønsten (2014), who showed parallel SWE measurements between snow pillows, snow scales,
and manual snow courses. With mitigating circumstances (e.g. snow drifting and scale issues), they concluded
that the measurement surface area had an impact on the measurement quality and that the Sommer scale
gave "promising results" but that further intercomparison was required.
One of the over-all objectives of the WMO-SPICE project is to assess the performance of automated
instrumentation for the measurement of snow, including snow on the ground (SoG). This is accomplished by
comparing the tested instruments to an established reference measurement. In total, fifteen countries are
participating in the WMO-SPICE project with about 20 intercomparison sites. Of these, 7 countries and 9
intercomparison sites are hosting SoG instrumentation. The instrumentation for WMO-SPICE has either been
provided by the instrument manufacturers or by the site hosts. For SoG, 13 different instruments are under
test with 9 measuring snow depth and 4 measuring SWE. The CS725 and the SSG1000 SWE instruments
examined here were installed at the Sodankylä (Finland), Caribou Creek (Canada) and Weissfluhjoch
(Switzerland) intercomparison sites (Fig. 1). To supplement the CS725 data collected for WMO-SPICE, data was
added from an additional CS725 instrument installed at the Fortress Mountain ski area in the Kananaskis region
of the Canadian Rocky Mountains.

# 2 Instrumentation and Methods

## 2.1 Campbell Scientific CS725

The CS725 (Fig. 2 left) is a passive gamma sensor developed by Hydro Québec in collaboration with Campbell Scientific (Canada) Corp. (Choquette et al., 2008; Martin et al., 2008). The instrument is installed above the snow surface and determines SWE by measuring naturally emitted gamma radiation from Potassium and Thallium sources in the soil that is attenuated by the snowpack. Each gamma ray detected by the sensor element is counted over a user defined period, the resulting distribution is compared to the distribution when there was no snow cover, and the difference is used to calculate SWE. The sensor field of view is approximately 120° resulting in a measurement area of approximately 80 m$^2$ when installed 3 m above the snowpack and with the collimator attached. The collimator serves to shield the instrument from gamma rays emitted from sources that are not in the target area. The effective range of the instrument is 0 to 600 mm w.e. with a measurement accuracy of +/- 15 mm w.e. from 0 to 300 mm w.e. and 15 % from 300 to 600 mm w.e. (Campbell Scientific CS725 Manual, https://s.campbellsci.com/documents/ca/manuals/cs725_man.pdf).

The two CS725 instruments for WMO-SPICE were both installed in October 2013 at Sodankylä, Finland, and Caribou Creek, Canada, and operated over the Northern Hemisphere winters of 2013/2014 and 2014/2015. Both instruments were mounted so that the bottom of the instrument was approximately 2 m above the ground and both were installed with the manufacturer provided collimator. Data was output every 6 hours. The instrument at Sodankylä was moved approximately 10 m during the summer of 2014 to avoid some buried cables in the measurement area, but any potential impact of the move are considered to be negligible because of the consistency in the snowpack at this site. The impact of spatial variability is addressed in Section 4.

The third CS725 used in this analysis was not a WMO-SPICE instrument, but was loaned to the University of Saskatchewan for testing and intercomparison by the instrument manufacturer. This instrument was installed

in a clearing near the Fortress Mountain ski resort in the Kananaskis Valley, Alberta, Canada. The CS725 was
mounted at a height of approximately 3.5 m above the ground. The distance to the trees around the
instrument was approximately 10 m from the centre of the instrument, , putting them outside of the response
area. Data collected by this instrument from October 2013 through June 2014 is used in this analysis. Like the
other CS725 instruments, SWE data was output every 6 hours.

## 7  2.2   Sommer SSG1000

The SSG1000 snow scale (Fig. 2 right) manufactured by Sommer Messtechnik, Austria, measures SWE through
the use of a weighing platform and load cells. Unlike the CS725, it makes a direct measurement of the weight
of the snowpack on top of the weighing platform and converts this weight to SWE. The entire platform consists
of 7 perforated panels, each 0.8 m x 1.2 m, that are attached to a frame and installed level with the surface of
the ground. The entire instrument surface is 2.8 m x 2.4 m (6.72 m$^2$) but only the centre panel is weighed by
the load cell. According to the manufacturer, the purpose of the larger surface surrounding the centre
measurement panel is to "stabilize" the overlying snowpack and prevent ice bridging
(http://www.sommer.at/en/products/snow-ice/snow-scales-ssg). The SSG1000, as tested for WMO-SPICE, has
a measurement range of 0 to 1000 mm w.e., and a manufacturer stated resolution and accuracy of 0.1 mm
w.e. and 0.3 % of full scale (3 mm w.e.), respectively.
The SSG1000 snow scales in this analysis were installed in the Sodankylä and Weissfluhjoch SPICE sites. The
Weissfluhjoch instrument was provided by the WSL Institute for Snow and Avalanche Research SLF. Data
collection from the instrument started in October 2013 and continued for the 2013/2014 and 2014/2015
Northern Hemisphere winters. The SSG1000 at Sodankylä was located in the North East quadrant of the SPICE
Field, approximately 22 m southeast of the original location of the CS725. At Weissfluhjoch, it is located in the
southwest corner of the instrument field. SWE observations from the instruments were recorded once per
minute during the two intercomparison seasons.
## 2.3  Reference SWE measurements
The reference SWE manual measurements for this intercomparison differed by site. All were bulk density snow
samples made with a snow sampling tube of a known diameter that has one end capable of penetrating and
cutting into the snowpack. The tube was inserted into the snowpack either down to the surface of the ground
or to a plate inserted into the snowpack, and the sample was extracted. Along with the sample, the depth of
the snowpack was also obtained. The sampled snow was then either bagged and weighed or was weighed
inside the tube using a cradle and balance. The snow sampler used in Canada is different than the tube used in
Finland and these differences, as well as any other differences in sampling technique, are described below.
At Caribou Creek, the reference SWE measurements were obtained using an ESC-30 snow tube with a 30 cm$^2$
cutting area. Farnes et al. (1983) and Goodison et al. (1987) show that the ESC-30, when used correctly and in
ideal conditions, has a mean measurement error of less than 0.5 % of the true SWE. Errors associated with
sampling in less than ideal conditions are discussed later. Bulk density samples at Caribou Creek were taken
just inside the response area of the CS725, bagged, and weighed. A 30 cm$^2$ sample from within the response
area is assumed to have a negligible impact on future sensor measurements considering the total sensor
response area is 80 m$^2$, but it was filled in with discarded snow when possible. These manual SWE
measurements were made about every two weeks in conjunction with a full 5 point snow course across the
Intercomparison Field and into the forest canopy on each side.
At Sodankylä, the reference SWE measurement was made using a Finnish bulk density sampling tube, with a
sampling area of 78.54 cm$^2$, and balance (Kuusisto, 1984) at roughly the same location in the Intercomparison
Field every two weeks. Only one sample was measured at a time. During the winter of 2013/2014, the bulk
density SWE sample was obtained approximately 12 m from the centre of the CS725 measurement area and
approximately 16 m from the centre of the SSG1000. In 2014/2015, after the CS725 was moved, the manual
sampling was done approximately 6 m from the CS725 measurement area and approximately 25 m from the
SSG1000.
An ESC-30 snow tube was used at the Fortress Mountain site. A full snow survey was conducted at the site
once per month, transitioning to bi-weekly during the ablation period. Although the actual snow survey course
was through the forested area, supplemental measurements were taken in the clearing where the
instrumentation is located. The distance between the sensor and the manual measurements was
approximately 10 m.

13       The manual SWE measurements at Weissfluhjoch were performed by-weekly on the SLF study plot using

a bulk density Aluminum sampling tube with a sampling area of 70 cm$^2$ and length of 55 cm. The weight was
measured with a cradle and balance (Jonas et al., 2009). . The distance between the sensor and manual snow
measurement varied from observation to observation as the location of the snow pit was relocated for each bi-
weekly measurement. The average distance was approximately 20 m.
2.4   Intercomparisons
The intercomparisons are not completely consistent amongst the four sites because of the different
instrumentation and manual methods for measuring reference SWE. At Sodankylä and Weissfluhjoch, the
sensors can both be compared with the manual SWE measurements made nearby, although the manual
measurements are not within the measuring area of either instrument. The timestamps of both instruments
were matched as closely as possible to the manual observation time. Since the CS725 only reports every 6

hours, the measurement output closest to the manual observation time was used for the intercomparison.

Since the SSG1000 reports every minute, no time adjustment was necessary. The same procedure was used to

compare the CS725 to the SSG1000. No SSG1000 was present at Caribou Creek or Fortress Mountain and no

CS725 sensors were installed at Weissfluhjoch.

For the CS725, which outputs a SWE value derived from both the Potassium (K) and Thallium (Tl) counts, the

manufacturer suggests that the output with the higher count is generally the most reliable. For Sodankylä, the

K/Tl ratio is always greater than 1 (varying from 3.5 to 8.0) indicating that the Potassium counts are greater

than the Thallium counts. For Caribou Creek, the ratio varies from 2.8 to 4.0. For Fortress Mountain, the ratio

varies from 0.3 to 8.5 but is above 1 approximately 70 % of the time. Therefore, the CS725 analysis is based on

the Potassium output although the statistics for Thallium are shown in parenthesis in Table 1. This will allow us

to determine if there were any obvious differences in the statistics related to the output derived from one

source or the other.

# 3   Results

## 3.1   CS725 vs. manual

The comparison between the CS725 measurements and the manual SWE observations are shown in Fig. 3 with

the Potassium output in red circles and the Thallium output in blue triangles. The black line in the figure

represents the 1:1 line and the error bars represent the manufacturer's stated sensor accuracy. Figure 4 shows

the time series of automated and manual SWE measurements. Figure 5 shows the difference between the

CS725 and the manual measurement (red) and the measured air temperature (blue) through the two seasons.

The regression analysis coefficients and summary statistics are listed in Table 1. The statistics are provided for

each individual season and for the two seasons combined. The statistics for the individual seasons are also

refined further to show results for the accumulation period (delineated from the ablation period by the timing
of maximum seasonal SWE). This will help to eliminate the effects of snow melt on both the manual
measurement and the various potential impacts on the CS725 measurement. These figures and tables are
further analyzed for each site in the following subsections.
### 3.1.1   Sodankylä
Throughout the intercomparison periods at Sodankylä, the CS725 overestimated SWE on average by 30 %
(mean relative bias or MRB) as compared to the manual measurements. From Table 1, the regression analysis
for the CS725 as compared to manual SWE over the entire season results in a slope ($\beta$) of 1.24 for 2013/2014
and 1.06 for 2014/2015. The difference in $\beta$ between the K and Tl outputs is small. The intercepts ($\varepsilon$) for the
entire seasons are 8.77 mm w.e. for 2013/2014 increasing to 26.9 mm w.e. for 2014/2015. This difference
might be in part a result of moving the instrument to a new location. The correlation coefficient, $r^2$, is 0.92 for
2013/2014 and 0.96 for 2014/2015. With the period of ablation eliminated from the analysis, the impact on $\beta$
and $\varepsilon$ are relatively small although the intercept $\varepsilon$ decreases almost 9 mm w.e. and 4 mm w.e. for the
respective seasons. The accumulation period $r^2$ increases to 0.97 and 0.99 for the 2013/2014 and 2014/2015
seasons respectively suggesting that more scatter is introduced into the relationship during the ablation
period. This is discussed further below.
Figure 4 (top) shows the time series for the 2013/2014 (left) and 2014/2015 (right) seasons at Sodankylä. In this
figure, the overestimation of the CS725 (red and blue lines) can be seen when compared to manual SWE (black
circles). In general, the instrument trends are the same as for the manual measurements with differences
between the measurements increasing after the start of the ablation periods and in January 2014 and
December 2014. Although it appears from Fig. 5 that the difference between the measurements is simply
increasing with time (or SWE amount), we believe that at least part of this increase is a result of melting in the
snowpack which occurs during some relatively warm days. In 2013/2014 (Fig. 5, left), a large increase in the
difference occurs after the > 0 °C temperatures in mid to late April. In 2014/2015 (Fig. 5, right), there is a
moderate increase after some > 0 °C temperatures in March but a much larger jump after the beginning of the
ablation period in April.
3.1.2   Caribou Creek
The comparison of the CS725 instrument and the manual SWE measurements made at Caribou Creek are
shown in Fig. 3 and summarized in Table 1. As with Sodankylä, the difference between the two sensor outputs
(Potassium vs. Thallium) is negligible. Also like Sodankylä, the CS725 at Caribou Creek consistently
overestimates total SWE such that the MRB is 35 %. However, the relationships between the instrument and
the manual SWE measurements are different than at Sodankylä. At Caribou Creek, the slopes of the regression
line, β, are less than 1 for all scenarios in Table 1 with the exception of the accumulation period in 2014/2015.
The intercepts (ε) are all larger than seen at Sodankylä, with the accumulation period in 2014/2015 being the
exception once again. The $r^2$ values range from 0.90 for the combined (2013/2014 and 2014/2015) data to 0.55
for the accumulation period in 2014/2015.
For both the 2013/2014 and 2014/2015 seasons, the time series for Caribou Creek (Fig. 4 middle) shows a rapid
increase in SWE in early winter related to heavier, wet snowfall events that most likely began as rain and
transitioned to snow. For 2013/2014, the CS725 time series generally follows the trend of the manual SWE
measurements with a large deviation developing mid- to late-March with the onset of seasonal ablation. Figure
5 (middle) shows the time series of the difference between the CS725 and manual SWE (red) and the
temperature time series (blue) for both seasons. In 2013/2014 (Fig. 5 middle left), there is an increase in the
difference that occurs in late January. This could be due to a melt period where temperatures at the site
exceeded 4 °C preceding the increase in the instrument bias. A much larger jump in the difference occurs mid-
March possibly due to significantly higher temperatures (exceeding 10 °C) earlier that month. In 2014/2015
(Fig. 5 right), the deviation between the measurements occurs earlier in the season (mid- to late-January)
coinciding with a January snow melt period characterized by above 0 °C air temperatures and high wind speeds
(not shown) that resulted in ice layers on top and within the snowpack (which make accurate manual SWE
measurements more difficult) and possibly infiltration of melt water into the frozen sandy soil. Differences
decrease after snowfall events in February only to increase again after the start of ablation in March.
In reaction to an observed offset after the 2013/2014 intercomparison season, soil moisture and temperature
probes were installed at the Caribou Creek site with the objective of correlating post-calibration, overwinter,
and ablation soil moisture changes with sensor offsets. The instruments were installed at three depths: 0 to 5
cm (vertically), 5 cm (horizontally) and 20 cm (horizontally). Unfortunately, the probes only measure liquid
water(volumetric water content, or VWC) so the analysis is mostly limited to when the soil temperatures (also
measured by the probe) are above 0 °C when we assume that most of the water in the soil is unfrozen.
Figure 6 shows the time series of soil moisture near the surface (0 to 5cm) along with the difference between
the CS725 and manual measurements (scaled by a factor of 100 for visualization) for the 2014/2015 season.
The red markers indicate when the soil temperature at this level is above 0 °C. It is easy to see from the time
series when the liquid soil moisture (near the surface) freezes in late fall resulting in a rapid drop in measured
VWC. Following the freezing of the near surface layer, which occurs 8 November 2014, the measured soil
moisture in this layer remains static until mid-March 2015 when a period of positive air temperatures (Fig. 5
middle right) raises the near surface soil temperatures above freezing, transitioning frozen soil moisture to
liquid and allowing for further infiltration of snow melt water into the sandy soil. The near surface (0 to 5 cm)
soil temperatures rose above freezing even with snow on the surface.  The snowpack was patchy (verified from
hourly photos) and shallow, and meltwater was likely percolating through the snow and into the top layers of
the soil.
The freezing of the 0 to 5 cm depths in early November is preceded by rain/snow events in late October that
are represented by the large jump in CS725 SWE shown in Fig. 4 (middle right) and confirmed with snow depth
measurements (not shown). During the transition from rain to snow and prior to the surface freezing, Fig. 6
shows fluctuations in near surface soil moisture (measured by the soil moisture sensor as volumetric water
content or VWC) related to the precipitation events in late October and early November. The soil moisture
calibration of the CS725 sensor was entered as a gravimetric water content (GWC) of 0.10, which can be
converted to VWC by multiplying by the specific gravity of the soil (Lambe and Whitman, 1969).  The specific
gravity of the loose sand near the surface at Caribou Creek was estimated to be 1.4 based on nearby
measurements taken during the BOREAS campaign (Anderson, 2000).  The increase in measured VWC from the
calibration value to 0.18 (GWC of 0.13)  prior to freezing has the potential to create a small but potentially
perpetuating offset of up to 3 mm w.e. in the CS725 SWE estimates and may explain at least some of the bias
shown by the instrument beginning in mid-December.
In addition to the offset in the CS725 SWE measurements that occurs at the beginning of the season, it was
anticipated that the rapid increase in the difference between the CS725 and Manual SWE at the end of January
2015 could also be attributed to a change in near surface soil moisture, as this was a time of mid-season snow
melt. However, a change in the liquid soil moisture during the melt period could not be detected by the soil
moisture sensors so it is unlikely that the increase in the instrument offset can be attributed to infiltration of
melt water into the sandy soil. A more plausible explanation are manual measurement errors that could result
from attempting to sample a complex snowpack containing ice layers in the pack or at the snow/soil interface.
Ice layers would have formed due to mid-season melt and re-freezing. The increase in the difference between
the manual measurement and CS725 in mid- to late-March could be a result of snow melt infiltrating into the
top layers of the sandy soil as the soil thaws or forming a basal ice layer (Woo et al., 1982; Lilbaek and
Pomeroy, 2008) on top of the soil. A corresponding spike in measured soil moisture during early spring snow
melt is shown in Fig. 6.

### 6    3.1.3   Fortress Mountain

The intercomparison of the CS725 instrument and the manual SWE measurements made at Fortress Mountain
are shown in Fig. 3 (bottom) and summarized in Table 1. Unlike the other two sites, the CS725 and manual SWE
measurements generally fall on the 1:1 line with no systematic overestimation (MRB < -5 %). This can also be
seen in the time series shown in Fig. 4 (bottom). The slope of the regression line is 0.88 with a small decrease
to 0.76 when excluding the ablation period. The intercept is 32.4 mm w.e. increasing to 84.4 mm w.e. when
excluding the ablation period. The $r^2$ is comparable to Sodankylä at 0.92 (increasing to 0.94 by excluding the
ablation period). It is unfortunate that the sample size is relatively small (n=8) but regardless, the instrument
compares quite well to the manual measurements at this site.

### 16    3.2   SSG1000 vs. Manual

The regression analysis for the SSG1000 intercomparisons is shown in Fig. 7 with the time series for both
seasons shown in Figure 8. The comparison statistics are in Table 2. This analysis, as for the CS725 above, is
organized by site.

### 21    3.2.1   Sodankylä

The SSG1000 regression analysis with the manual SWE measurements shown in Fig. 7 (top) and summarized in
Table 2 has an $r^2$ for the entire 2014/2015 period of 0.99 but is only 0.84 for the 2013/2014 period. However,
the SWE data from the SSG1000 is not available for the ablation period in 2014/2015 due to an instrument
malfunction. To have a consistent intercomparison for the two seasons, the ablation period (post maximum
SWE) was removed from the 2013/2014 period and the $r^2$ becomes 0.97, very similar to 2014/2015. Combining
the two seasons, the slope of the regression, β, becomes 0.99 with an offset ε of -7.27 mm w.e. with an $r^2$ of
0.88. The MRB for the two seasons combined is -11 %.
The time series of these data are shown in Fig. 8 (top) for both the 2013/2014 (left) and 2014/2015 (right)
seasons. For both seasons, the sensor measurements track quite well with the manual measurements. The
outliers that appear in Fig. 7 (top) can also be seen in the 2013/2014 time series (Fig. 8 top left) beginning mid-
way through the ablation period. It is unknown if this occurs during the 2014/2015 ablation period because the
data are missing due to a sensor failure caused by water damage to the electronics (an issue later remedied by
the manufacturer).
### 3.2.2   Weissfluhjoch
The regression analysis for the SSG1000 and the manual SWE measurements is shown in Fig. 7 (bottom) with
the time series in Fig. 8 (bottom). This alpine site has a much deeper snowpack than either Caribou Creek or
Sodankylä but comparable to Fortress Mountain, which unfortunately did not have concurrent SSG1000
measurements. The $r^2$ for both seasons is quite high at 0.97, similar to the accumulation period
intercomparison at Sodankylä, but β is less (0.72 and 0.82) and ε is much higher (91.7 and 79.0 mm w.e.) for
both seasons (2013/2014 and 2014/2015). The outliers are obvious in Fig. 8 (bottom) when the manual SWE
measurements are substantially higher than the sensor measurements. Unlike Sodankylä, these outliers mostly
occur before maximum seasonal SWE, which is why we don't break the season down as we do with Sodankylä.
They are, however, likely a result of sensor bridging which is discussed more in Section 4. There are also
outliers that occur late in the ablation periods, where the sensor substantially overestimates SWE, and these
are perhaps due to issues with the manual sampling of a complex (melting or melting/refreezing) snowpack.
When combining the two seasons, the resulting low MRB of 8% (for combined seasons) is somewhat surprising
given the obvious outliers.  Perhaps the low combined MRB is a reflection of errors in one season
compensating the errors in the other season.

## 3.3 CS725 vs. SSG1000

The intercomparison with manual measurements for both the CS725 and the SSG1000 are suggesting that the
agreements are the most favourable during accumulation rather than during ablation. Figure 7 shows the
relationship between the CS725 and the SSG1000 for both seasons at Sodankylä with the 2014/2015 season
shown in red circles and the 2013/2014 season shown in blue dots (changing to blue triangles at the
approximate onset of ablation). The relationship for both years appears to be linear up to the time where
maximum SWE is reached. At the onset of ablation, the relationship between the instruments (shown by the
blue triangles) deviates substantially from linear. This is confirmed by Table 3 which shows a higher $r^2$ when the
2013/2014 ablation period is not included in the analysis. This analysis could only be completed for the
2013/2014 season since the sensor data is missing for the 2014/2015 ablation period due to malfunction.

## 3.4 Sensor Reliability

Data quality control metrics for the CS725 sensors at each of the two SPICE sites demonstrated that the
instruments performed at a high level of reliability, such that over 95 % of the sensor measurements were
usable for intercomparison. No malfunctions were noted and no maintenance was required at any of the sites.
For the SSG1000, data quality control metrics show that the sensors performed reliably during the
accumulation periods but malfunctioned at Sodankylä late in the spring of 2014 and again early spring of 2015.
At Weissfluhjoch, 99% of the 1-minute data was usable for intercomparison. At Sodankylä, the malfunctions
resulted in only 83 % and 67 % of the 1-minute data, for the 2013/2014 and 2014/2015 seasons respectively,
being available for intercomparison. The sensor malfunctions at Sodankylä were determined to be related to
water damage to the electronics. Other than this, no other malfunctions were reported or maintenance
required during the intercomparison.
## 4   Discussion
The regression analysis between the CS725 and the manual SWE measurements resulted in $r^2$ values ranging
from 0.55 to 0.99, depending on site and season.  Combined season $r^2$ values ranged from 0.90 to 0.92.
Although generally lower than the correlations of 0.99 reported for intercomparisons with other instruments
by Wright et al. (2011), our correlations (averaged by season) are similar to the $r^2$ of 0.83 that they reported for
snow tube measurements. The (combined season) bias shown here, which was between 30% and 35%, is
substantially higher than the 18% reported by Choquette et al. (2008). The exception to this is the CS725 at
Fortress Mountain which has a mean negative bias less than 5% when compared to the manual measurements.
Besides the maximum SWE, the two major differences that Fortress Mountain has from Caribou Creek and
Sodankylä are the soil and the topography. Soils at the Fortress Mountain site have higher clay and loam
content, overlain with a layer of organics, and generally remain frozen and saturated for the duration of the
winter. These, combined with the sloping terrain and faster meltwater runoff via drainage channels, likely
minimizes the change in soil moisture during the transition seasons and thereby minimizes potential offsets in
the CS725 measurements. Furthermore, the correlations for the CS725 for Caribou Creek are substantially
lower than for Sodankylä and Fortress Mountain. This could be for several reasons. The spatial and seasonal
variability are quite high at Caribou Creek and the sample size is low. This is especially the case for 2014/2015
where sample size is small due to a shorter and more variable winter where melt and re-freeze occurred
several times over the course of the season (Fig. 5 middle right). Melting and re-freezing generally makes the
manual SWE measurements more difficult and prone to error, creates basal ice, and results in higher spatial
variability. Eliminating the ablation period improved the comparison statistics for 2013/2014 but made the
statistics for 2014/2015 much worse due to the reduced sample size. Potential sources of error in the CS725
intercomparison are discussed further in the following sections.
The SSG1000 was quite highly correlated with the manual SWE measurements at both Sodankylä and
Weissfluhjoch with $r^2$ values as high as 0.99 at Sodankylä (when excluding the ablation period) and 0.97 at
Weissfluhjoch. However, when the ablation period is included in the intercomparison for 2013/2014 at
Sodankylä (it is not present in 2014/2015 at Sodankylä due to sensor malfunction), the $r^2$ drops to 0.84. The
more significant result at Sodankylä is the smaller MRB as compared to the CS725, which is -2% to -15%
(depending on the exclusion of ablation). The magnitude of the MRB is similar at Weissfluhjoch but the bias
here is a positive 8%. This is surprising considering the many occurrences of negative sensor bias (as seen in
Figure 8 bottom) but these negative outliers are balanced by some large (albeit inconspicuous) positive outliers
at the end of the ablation periods. The outliers for Sodankylä in Fig. 7 (top) occur during the ablation period in
late April to May 2014 but it is difficult to ascertain if the errors are related to the instrument or to the manual
measurement. The most likely explanation is that these are related to the occurrence of bridging. Bridging is
also suspected as the cause of the pre-ablation outliers at Weissfluhjoch since the sensor seems to agree quite
well with the manual measurements up to mid-March and early-April for both seasons. An intercomparison
with a collocated snow pillow (not shown here) suggests a similar albeit smaller negative bias during the same
period. Errors associated with bridging are discussed further in this section.
The CS725 and SSG1000 measurements at Sodankylä correlate very well with each other showing correlations
as high a 0.99 when excluding the ablation periods. The key result here, as shown in Fig. 9, is the deviation from
this linear correlation at the onset of melt in the 2013/2014 season. Although some of this deviation can be
blamed on differential melting at the site, we attribute a large portion of the deviation to the different
measurement principles of the sensors. At the onset of melt and the ripening of the snowpack, meltwater
drains out of the snowpack towards the ground surface. Once reaching the surface, the meltwater can pool
and re-freeze (potentially forming a basal layer of ice), runoff from the measurement area, or infiltrate into the
soil. Due to the flat measurement area and the sandy soil at Sodankylä, runoff is unlikely; therefore the
meltwater is either infiltrating into the sandy soil or re-freezing at the surface. Either way, the same meltwater
is likely draining through and away from the measurement plate of the SSG1000 and therefore no longer being
measured as SWE in the snowpack. However, this meltwater, whether infiltrated into the top layer of the sandy
soil or pooling at the surface, is still being registered by the CS725 as SWE. This contributes to the
overestimation of SWE by the CS725 as compared to the SSG1000 and to the non-linearity of the
intercomparison shown in Fig. 9 after ablation. Also, this meltwater is either difficult or impossible to include in
a snow tube sample, increasing the bias between the CS725 and the manual measurements.
## 4.1   Sources of error
There are several possible sources of error that affect both the automated and manual SWE measurements.
They are discussed and analyzed for each instrument/method in this section.
### 4.1.1   Soil moisture (CS725)
A potential source of error for the CS725 can arise from a poor pre-snowpack soil moisture calibration or a
large post-calibration change in soil moisture prior to the freezing of the ground surface. Overwinter soil
moisture changes (Gray et al., 1985) or infiltration of snowmelt water into soils (Gray et al., 2001) could also
result in deviation between the manual and CS725 SWE measurements. Since the CS725 calculation of SWE is
based on gamma ray counts during wet and dry periods with no snow cover, incorrect measurements or faulty
assumptions with respect to the soil moisture calibrations could result in a sensor offset. Furthermore, if soil
moisture levels change significantly prior to freeze-up, during winter, or during ablation, then the SWE
estimates derived from the sensor are less reliable. The approximate error associated with an inaccurate
gravimetric soil moisture calibration, as provided by the manufacturer, is roughly 10 mm w.e. of SWE for a 0.10
change in GWC. Figure 6 shows an increase in soil moisture at Caribou Creek up to a VWC of 0.18 (GWC of 0.13)
prior to freeze-up in the fall of 2014, an increase of 0.03 GWC and approximately 3 mm w.e. The resulting
calibration offset could explain up to 30 % of the early season difference between the instrument and the
manual measurement shown in Fig. 4 (middle right) and Fig. 5 (middle right). This calibration issue would then
perpetuate through the winter period and grow with any additional infiltration into the soil beneath the
snowpack. It is unfortunate that this same soil moisture and soil temperature data are not available for
Sodankylä or for the first season at Caribou Creek as this would have provided some verification for the
calibration offset.
From Fig. 6, there appears to be a coinciding jump in the CS725 bias and the jump in soil moisture (due to
above freezing soil temperatures and infiltration) in the spring of 2015 at Caribou Creek. Although the bias is
not as large as that seen in mid-winter, it is a significant increase of approximately 10 mm w.e. for each of the
final two intercomparison points in mid-March and early-April. Much of this 20 mm w.e. increase could be
explained by a corresponding increase in soil moisture from 0.18 VWC (0.13 GWC; estimated at freeze up) to
0.45 VWC (0.32 GWC; spike at thaw) or approximately 19 mm w.e., assuming that the CS725 is interpreting this
near surface soil moisture as SWE.
There is some ambiguity in the soil moisture results because the soil moisture sensors are incapable of
measuring moisture content below 0° C and because this is not the only source of error.  However, we think
that these soil measurements are useful for explaining at least some of the offsets seen between the sensor
and the manual SWE measurements, especially during the transition periods. More work is needed on these
linkages before a reliable sensor adjustment can be derived.

### 4.1.2 Ice bridging (SSG1000)

Ice bridging is a known issue affecting SWE measurements that are made by weight, such as snow pillows or
the snow scale (e.g. Engeset et al., 2000). Bridging typically occurs when air temperature reaches 0 °C and then
cools creating a melt-refreeze crust layer on the snow surface. This layer is very hard and supports the weight
of the snow, thus causing an underestimate of measured SWE with further accumulation on the surface.
Probable bridging situations can be seen in Fig. 7 both at Sodankylä and at Weissfluhjoch. At Sodankylä, in
December 2013, March 2014 and February-March 2015, the SWE values measured by the SSG1000 do not
increase as quickly as the manual measurements. At the same time, air temperature first goes above 0 °C and
then cools to as low as -30 °C creating perfect conditions for ice bridging. At Weissfluhjoch, the cause of
potential ice bridging is not so obvious, but it is difficult to explain the differences between manual and
SSG1000 measurements otherwise. The snowpack was homogeneous (verified with terrestrial laser scans) and
even though a co-located snow pillow (not shown here) showed some underestimation compared to the
manual measurements, the underestimation was much smaller than by the SSG1000. However, snow pillows
have been found to be less prone to ice bridging issues due to their larger surface area (Beaumont, 1966;
Tollan, 1970).  A more comprehensive description of the physical processes that cause measurement errors in
SWE pressure sensors can be found in Johnson (2004).

### 4.1.3 Snow spatial variability

Another potential source of error in this analysis is due to the spatial variability at the intercomparison sites
impacting the relative SWE between the sensor and manual measurement locations. At Sodankylä, the
maximum distance between the sensors and the manual SWE measurements was 12 m for the CS725 (6 m
after the move prior to the 2014/2015 season) and 25 m (16 m in 2013/2014) for the SSG1000. Unfortunately,
only one SWE measurement is made at the intercomparison site, but generally the spatial variability is low with

snow depth exhibiting a coefficient of variation (COV) under 6 % (with a maximum snow depth of just over 80

cm). Therefore, the impact of spatial variability in SWE, even with a 25 m separation, is likely quite small for

most of the season. However, both webcam photos and snow depth measurements provide evidence that

snow melt rates during ablation vary across the site, largely dependent on exposure. Manual snow depth

measurements suggest that spatial differences in the area around the SWE measurements are small and are

perhaps as high as 4 cm in mid-April of 2014 and less in mid-April of 2015. These differences obviously account

for very little of the late season SWE deviation shown in Fig. 5 (top). This also suggests that the CS725 move

prior to the 2014/2015 season had a low impact on sensor bias from one season to the next.

Caribou Creek, with maximum snow depths of 56 cm and 41 cm for the two consecutive seasons, exhibits a

much higher spatial variability. Here, COV is about 15 % (19 %) at peak snow depth but increases to 30 % (90 %)

during ablation for 2013/2014 (2014/2015). With a full 5 point snow course performed here, mean SWE

maximum is approximately 125 mm w.e. in 2013/2014 and 75 mm w.e. in 2014/2015 with COV very similar to

those shown for snow depth. The manual measurement used in the intercomparison is made just inside the

measurement area of the sensor, approximately 5 m from the centre. Although relatively close, the higher

spatial variability could result in a spatial bias, especially during ablation. For example, in 2013/2014, we

estimate SWE to increase across the sensor measurement area by approximately 10 mm w.e. in late April due

to differential melting as a result of exposure. With the manual measurement closer to the lower SWE estimate

in the sensor measurement area, up to 25 % of the difference in SWE between the sensor and the manual

measurement (as shown in Fig. 5 middle left) could be explained.

The spatial variability is not assessed for Fortress Mountain or Weissfluhjoch.

### 4.1.4 Experiment design

Some aspects of the design of the SWE intercomparison are less than ideal and often were a result of compromise amongst the overall SPICE objectives, site host resources, and nationally accepted practices. These compromises potentially contribute to some ambiguity of the study results and this commentary could form the basis for recommendations on the design of future SWE sensor intercomparisons.

Ideally, the manual reference at each site should have been identical using the same sampling equipment at a prescribed offset distance from each SWE sensor. Rather, each site host used their nationally accepted method of sampling SWE (as described in Section 2.3). Distances between the manual SWE measurement and the sensor varied from 5 m to 25 m, depending on site, but perhaps more significantly, the variation within the sensor measurement area (especially for the CS725) was not properly assessed. This could certainly have been a factor at Caribou Creek but the intense sampling within the measurement area of the sensor would have caused too much disturbance and impacted sensor measurements. Also, increased frequency (i.e. weekly) of manual measurements is desirable especially after significant changes in the snowpack, albeit at the risk of disturbance. In the future, manual observers should pay special attention to the existence of basal ice layers which may have an impact on the overall accuracy of the manual SWE estimate.

Another ideal situation would have been the co-location of both SWE sensor types at each site. This, in combination with soil moisture and temperature sensors within the measurement area of the CS725 sensors, would have provided additional information for the assessment of sensor bias. Another good addition would be the automated and high frequency measurement of snow depth within the sensor measurement areas to provide an indicator of snow density and melt rates and perhaps an indicator of snow bridging on the weighing SWE sensors.

### 4.1.5 Manual SWE measurements

As noted above, the manual SWE measurements differed by site, the exception being Caribou Creek and Fortress Mountain that both used the ESC-30 snow tube and bagged and weighed the sample. We won't comment further on possible bias associated with different samplers (Farnes et al., 1983; Goodison et al., 1987), as these are generally small as compared to the differences in the measurements shown in these results. We do, however, want to address possible errors associated with the manual measurement of a complex snowpack (i.e. a snowpack with ice layers or during melt), especially with a snow tube.

During the intercomparison, both Caribou Creek and Sodankylä experienced several freeze and thaw cycles over the course of the winter (as seen in Fig. 5 top and middle) but one was especially pronounced at Caribou Creek during mid- to late-January 2015 (Fig. 5 middle right). The result of freeze/thaw is usually a "crusty" snowpack with several ice layers. In general, these characteristics make a snowpack difficult to sample with a snow tube as the tube cutters need to cut through multiple ice layers without snow escaping from the bottom of the tube (Powell, 1987; Sturm et al., 2010). It is anticipated that even an expert user will have difficulties obtaining an accurate sample in these conditions, exacerbated even more by the shallow pack found at Caribou Creek in 2014/2015. It is difficult even at the time of the sample to estimate measurement error, but it could easily result in a 5 to 10 % underestimate of SWE.  Sturm et al. (2010) reported an average underestimate from a snow tube of 7.1% as compared to layer-integrated snow pit measurements. Although this may explain some of the bias in the CS725 measurements, especially at Caribou Creek, it is countered by the relatively good agreement between the manual and SSG1000 measurements for Sodankylä. However, mid-winter melting could also result in basal ice as the meltwater percolates through the snow and refreezes at the surface (providing that the surface is below 0 °C) or in the top layer of the sandy substrate. Not only would this ice layer be difficult to measure with a snow tube (which is difficult to cut through and often results in an underestimate), the meltwater may drain off of the SSG1000 measurement surface and be underestimated by

1 that measurement as well. This may partially explain the often (but sometimes inconsistent) increase in sensor

2 bias shown by manual SWE measurements following mid-winter freeze/thaw cycles in Fig. 5 (top and middle).

3 Unfortunately, the observer's notes did not indicate when a basal ice layer was observed so much of this is

4 speculation.

6 During ablation, measures were taken to sample the snowpack before it ripened but this could not always be

7 accomplished due to travel time to the site (especially for Caribou Creek). Because the sample was bagged and

8 weighed rather than weighed in the tube, a wet sample would experience some errors because of the bagging

9 process (liquid water or sticky snow left in the tube) and result in an underestimate of SWE (perhaps 5 % as a

10 rough estimate).

## 5 Summary and Conclusions

13 Two automated SWE sensors were tested at three WMO-SPICE sites (Sodankylä, Weissfluhjoch and Caribou

14 Creek) and at one additional Canadian site (Fortress Mountain) during the WMO-SPICE intercomparison

15 (Northern Hemisphere) winters of 2013/2014 and 2014/2015. Instrument measurements were compared to

16 periodic manual measurements of SWE at the sites and cross referenced with ancillary measurements of air

17 temperature and soil moisture and soil temperature (at Caribou Creek) to try to determine causality for some

18 of the bias seen in the intercomparison. The objective was not necessarily to determine which instrument

19 makes the most accurate measurement, but to inform users of potential measurement issues that may

20 influence their data interpretation.

22 Intercomparison results for the CS725 show that it overestimates SWE on average by 30 % and 35 % at

23 Sodankylä and Caribou Creek respectively with combined season correlations ($r^2$) of 0.92 at Sodankylä and 0.90

at Caribou Creek. Interseasonal variability in both the MRB and the correlations were higher at Caribou Creek,
the differences attributed to smaller sample sizes, higher spatial variability of SWE, and ice layers in the
snowpack. Offsets were generally higher at Caribou Creek which could be indicative of an inaccurate soil
moisture calibration of the instrument, a change in soil moisture relative to the calibration prior to or after the
soil freezing, or sampling errors in the manual SWE measurement due to a more complex snowpack.
Correlations at Fortress Mountain are also quite high over the single intercomparison season ($r^2$=0.92) with a
mean negative bias of approximately 5 %, which is more comparable to the results of Wright et al. (2011) in
similar conditions. At the two sandy SPICE sites, the agreement between the CS725 and the manual SWE
measurements are generally better prior to the start of seasonal ablation. We believe this occurs largely
because of early spring melt percolating through the snowpack and either forming a basal ice layer or
infiltrating into the sandy substrate. Either way, this water is difficult or impossible to measure with a snow
tube. However, because this water continues to attenuate the gamma radiation signal detected by the CS725,
the sensor still interprets this water as SWE and therefore appears to overestimate as compared to the manual
measurements. Seasonal ablation has no significant impact on the agreement at Fortress Mountain due to
saturated frozen soils that restrict infiltration and a mild slope that promotes runoff of meltwater from the site.
The SSG1000 at both Sodankylä and Weissfluhjoch, compared quite well to the manual SWE measurements
showing mean biases of -11 % and 8% at the respective sites. It did, however, experience some technical issues
at Sodankylä early in the 2014/2015 snowmelt period which limited the intercomparison for that season. The
correlations were quite high with the combined season $r^2$ ranging from 0.88 at Sodankylä to 0.96 at
Weissfluhjoch. Many of the outliers in the SSG1000 intercomparisons are most likely due to bridging of the
snowpack on the weighing plate but we also have to consider errors related to the manual measurements and
other processes going on at the snow-soil-sensor interface (as outlined in Johnson, 2004). At Weissfluhjoch,
these outlier events occurred prior to maximum seasonal SWE while at Sodankylä they occurred during
ablation. Removing the ablation period in the 2013/2014 Sodankylä data resulted in a substantial increase in $r^2$
from 0.84 to 0.97.
The SSG1000 correlated very well with the CS725 at Sodankylä during the accumulation period. Although the
overestimation of SWE by the CS725 is quite apparent when compared against the SSG1000, the accumulation
period $r^2$ was 0.98 and 0.99 for the two respective seasons. Intercomparison of the two sensors clearly shows
how the overestimation of SWE by the CS725 increases at the onset of ablation in March/April of the
2013/2014 season. Independent of the manual measurements, this indicates that the deviation of the CS725
from manual SWE during ablation is most likely instrument related and a result of the CS725 misinterpreting
the melt water infiltrated into the sandy soils as SWE.
When comparing SWE instruments to a manual reference, there are several considerations that must be made
that ultimately impact the interpretation of the results. We know that the manual measurements of SWE are
not free of error. Experience proves that making a snow tube bulk density sample in a snowpack containing ice
layers or during melt is difficult and inherently prone to errors. We also have to consider the spatial variability
of the snow that we are sampling as the CS725 (and the SSG1000 to a lesser degree) have a much larger
measurement area than the manual point sample. Taking this and the technical capabilities of the instruments
into consideration, both sensors have high correlations (generally higher than 0.90, Caribou Creek being the
exception) with the manual reference measurements. We have identified that the SSG1000 has had some
technical issues during snowmelt but are satisfied that these issues can be overcome with some installation
modifications. The SSG1000 may also underestimate SWE on occasion due to bridging so users need to be
aware of this potential error. We have identified the potential for the CS725 measurements to be
misinterpreted, especially when deployed over sandy soils and during melting conditions.   Although more
verification work is required on the impact of soil moisture change on the CS725 bias, aggregating sub-surface
moisture in the SWE estimate could potentially be useful from a hydrological perspective as it ultimately
impacts the amount of water available for runoff. Nevertheless, it is recommended to co-locate the CS725 with
ancillary measurements of soil moisture, soil temperature, and snow depth to guide the user in interpreting
the data.
# Acknowledgements and Disclaimers
Many of the results presented in this work were obtained as part of the Solid Precipitation Intercomparison
Experiment (SPICE) conducted on behalf of the World Meteorological Organization (WMO) Commission for
Instruments and Methods of Observation (CIMO). The analysis and views described herein are those of the
authors at this time, and do not necessarily represent the official outcome of WMO-SPICE. Mention of
commercial companies or products is solely for the purposes of information and assessment within the scope
of the present work, and does not constitute a commercial endorsement of any instrument or instrument
manufacturer by the authors or the WMO. We wish to thank the WSL Institute for Snow and Avalanche
Research SLF for kindly providing the SSG1000 and manual SWE measurements from Weissfluhjoch as well as
the countless other contributors to SPICE who helped to make the project a success. We would like to express
our appreciation for the effort that the reviewers and the special issue editor provided to help us improve this
manuscript with a special thanks to Dr. Charles Fierz (WSL-SLF, Davos) who provided a very thorough review
with substantial and helpful feedback.

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

## 1 Tables

Table 1: Regression coefficients and other statistical measures for the multi-season intercomparison of the
CS725 with manual SWE at Sodankylä, Caribou Creek and Fortress Mountain (where β and ε are the slope and
intercept of the regression line). Values inside and outside of the parenthesis represent Thallium and
Potassium output, respectively, from the sensor. "Accumulation" indicates that data occurring after maximum
seasonal SWE is omitted from the analysis. "Combined" indicates that data from both seasons are included,
and n represents the sample size.

| Site | Season | β | ε mm w.e. | $r^2$ | RMSE mm w.e. | Mean Relative Bias % | n |
|---|---|---|---|---|---|---|---|
| Sodankylä | 2013/2014 | 1.24(1.27) | 8.77(3.17) | 0.92(0.92) | 43.0(42.2) | 30.1 | 17 |
| | 2013/2014 (Accumulation) | 1.24(1.28) | 0.0123(-6.63) | 0.97(0.97) | 35.6(33.9) | 24.6 | 13 |
| | 2014/2015 | 1.06(1.13) | 26.9(24.2) | 0.96(0.96) | 36.6(42.2) | 30.9 | 13 |
| | 2014/2015 (Accumulation) | 1.05(1.12) | 23.3(20.2) | 0.99(0.99) | 30.0(35.7) | 28.1 | 10 |
| | Combined | 1.16(1.21) | 16.8(11.9) | 0.92(0.92) | 40.3(42.2) | 30.4 | 30 |
| Caribou Creek | 2013/2014 | 0.783(0.764) | 40.6(46.9) | 0.78(0.72) | 22.8(27.5) | 22.2 | 12 |
| | 2013/2014 (Accumulation) | 0.982(0.997) | 17.7(20.2) | 0.79(0.75) | 18.0(22.2) | 15.4 | 9 |
| | 2014/2015 | 0.849(0.849) | 27.1(30.4) | 0.77(0.71) | 23.6(27.4) | 63.0 | 7 |
| | 2014/2015 (Accumulation) | 1.12(1.31) | -8.38(-14.5) | 0.55(0.60) | 25.4(29.5) | 42.4 | 4 |
| | Combined | 0.904(0.911) | 27.5(31.0) | 0.90(0.87) | 23.1(27.4) | 34.6 | 19 |
| Fortress Mountain | 2013/2014 | 0.881 | 32.4 | 0.92 | 48.0 | -4.5 | 8 |
| | 2013/2014 (Accumulation) | 0.764 | 84.4 | 0.94 | 56.0 | -3.6 | 5 |

Table 2: Regression coefficients and other statistical measures for the multi-season intercomparison of the
SSG1000 with manual SWE at Sodankylä and Weissfluhjoch (where β and ε are the slope and intercept of the
regression line).  "Combined" indicates that data from both seasons are included and n indicates the sample
size.

| Site | Season | β | ε mm w.e. | $r^2$ | RMSE mm w.e. | Mean Relative Bias % | n |
|------|--------|---|-----------|-------|--------------|----------------------|---|
| Sodankylä | 2013/2014 | 1.05 | -15.5 | 0.84 | 24.2 | -15.1 | 17 |
|  | 2014/2015 | 0.92 | 5.5 | 0.99 | 7.9 | -2.3 | 10 |
|  | Combined | 0.99 | -7.3 | 0.88 | 19.8 | -10.8 | 27 |
| Weissfluhjoch | 2013/2014 | 0.72 | 91.7 | 0.97 | 55.5 | 4.2 | 14 |
|  | 2014/2015 | 0.82 | 79.0 | 0.97 | 58.6 | 11.3 | 17 |
|  | Combined | 0.79 | 77.2 | 0.96 | 57.2 | 8.1 | 31 |

Table 3: Regression coefficients for the multi-season intercomparison of the CS725 with the SSG1000 SWE
measurements at Sodankylä (where β and ε are the slope and intercept of the regression line). "Accumulation"
indicates that data occurring after maximum seasonal SWE is omitted from the analysis.

| Season | β | ε mm w.e. | $r^2$ |
|--------|---|-----------|-------|
| 2013/2014 | 1.20 | 15.7 | 0.90 |
| 2013/2014 (Accumulation) | 1.24 | 4.29 | 0.98 |
| 2014/2015 | 1.19 | 11.9 | 0.99 |

# Figures

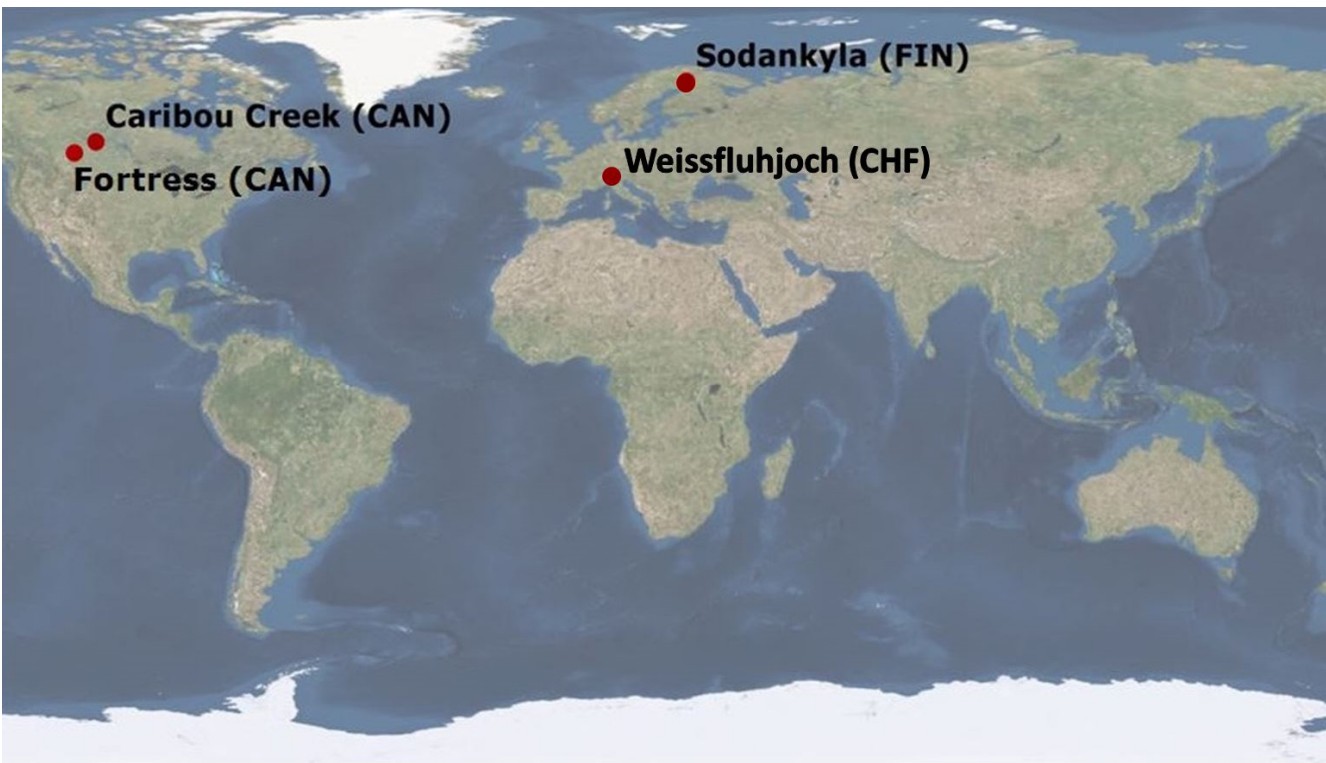

Figure 1: Location of the CS725 (Sodankylä, Caribou Creek, Fortress Mountain) and SSG1000 (Sodankylä and Weissfluhjoch) instrument intercomparisons.

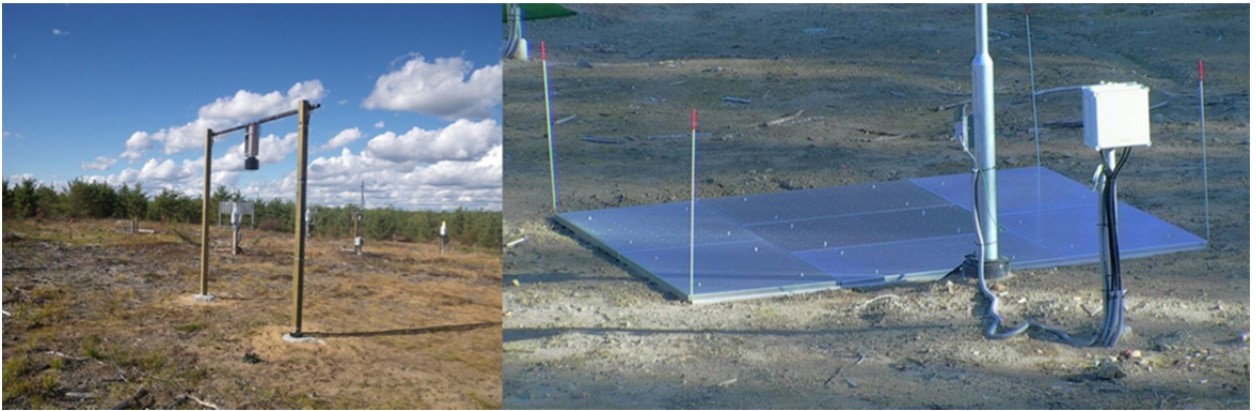

Figure 2: The Campbell Scientific CS725 (left) installed at Caribou Creek and the Sommer Messtechnik SSG1000 (right) installed at Sodankylä.

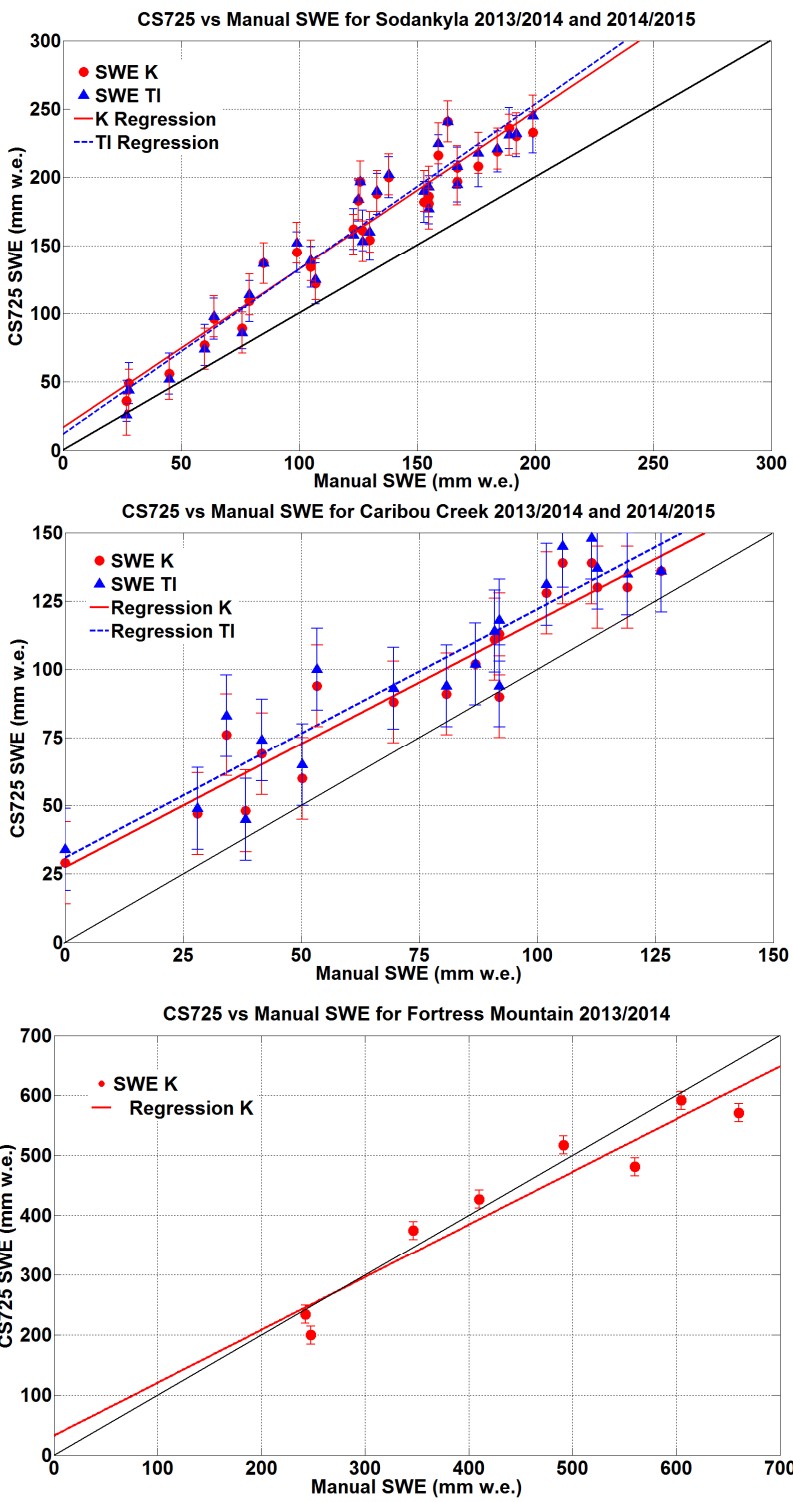

1    Figure 3: CS725 vs Manual SWE for Sodankylä (top) and Caribou Creek (middle) for the 2013/2014 and
2    2014/2015 seasons and Fortress Mountain (bottom) for the 2013/2014 season. Potassium output in red and
3    Thallium output in blue. Black line is 1:1. Error bars represent manufacturer's stated sensor accuracy.

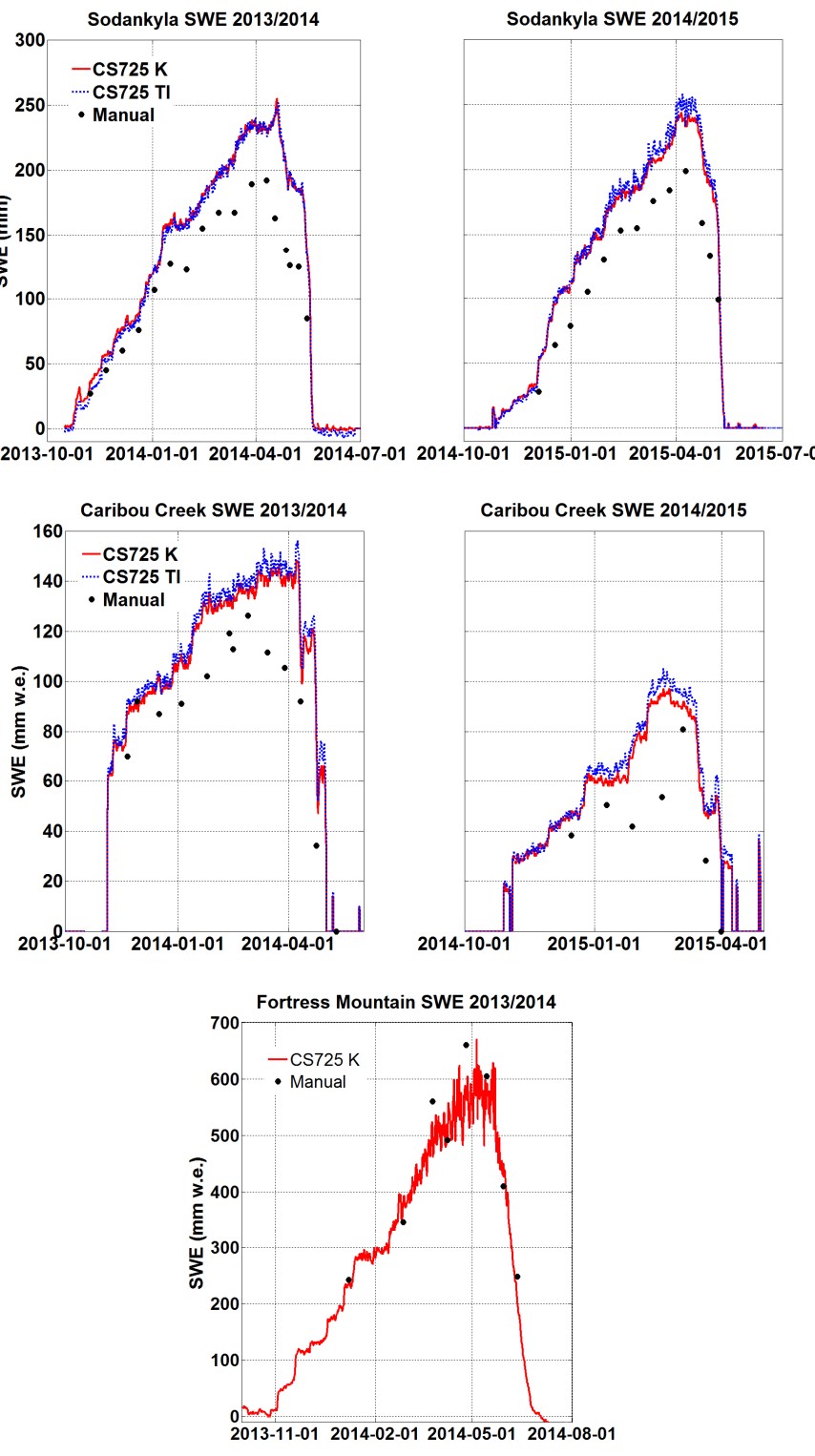

Figure 4: Time series of the CS725 SWE sensors and manual SWE measurements at Caribou Creek (top), Sodankylä (middle) for the 2013/2014 (left) and 2014/2015 (right) seasons, and Fortress Mountain (bottom) for the 2013/2014 season.

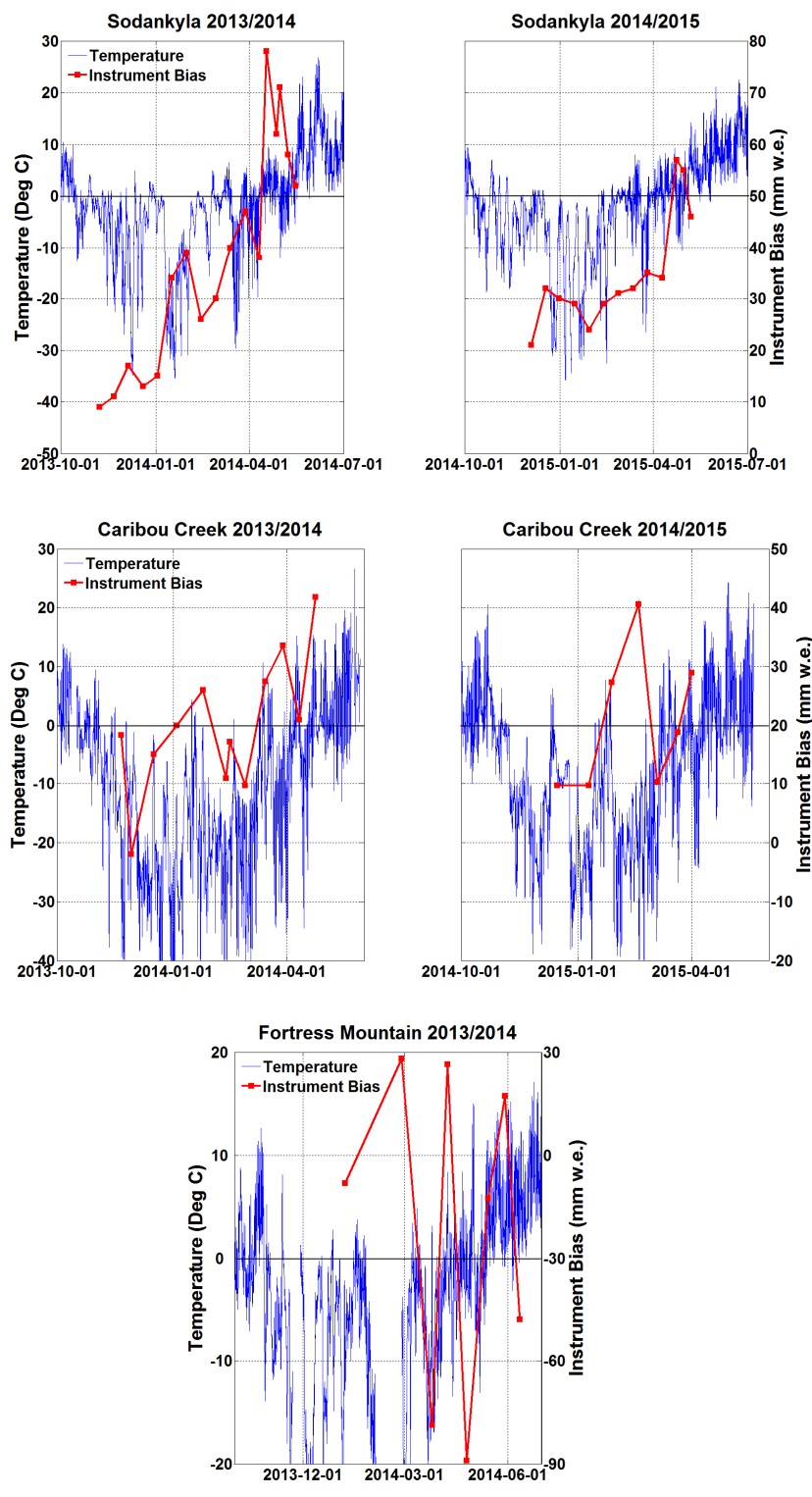

Figure 5: Time series of 1.5 m air temperature (blue, left axis) and difference between CS725 and manual measurements (red, right axis) at Sodankylä (top) and Caribou Creek (middle) for the 2013/2014 (left) and 2014/2015 (right) seasons, and at Fortress Mountain (bottom) for the 2013/2014 season.

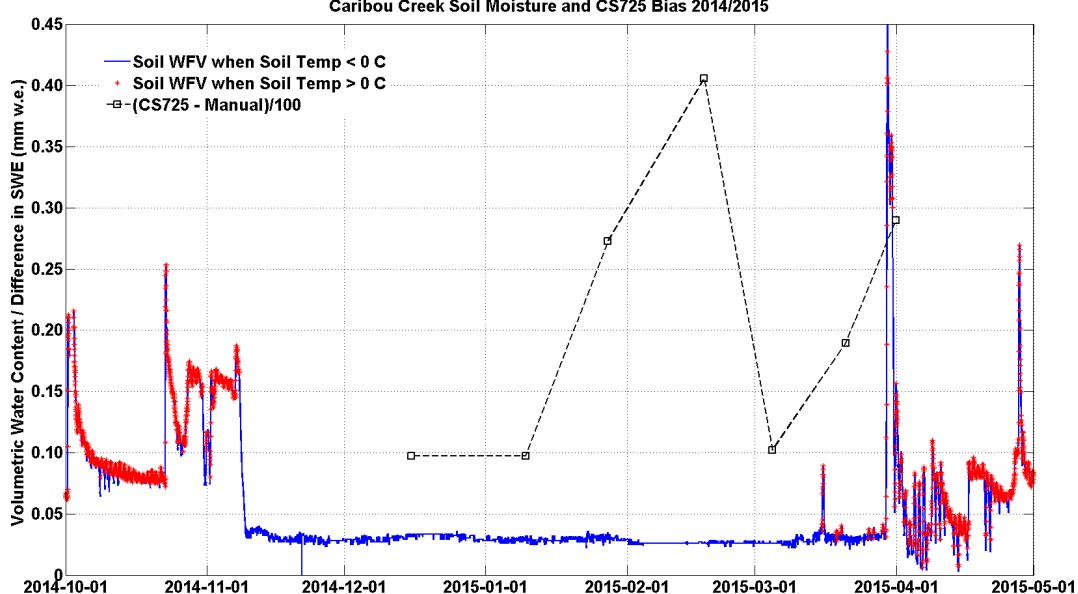

Figure 4: Time series of near surface (0 to 5 cm) soil moisture (volumetric water content, blue) and the
difference between the CS725 and manual measurements (dashed line and black boxes) at Caribou Creek for
the 2014/2015 season. Red markers show where near surface soil temperatures are above 0°C.

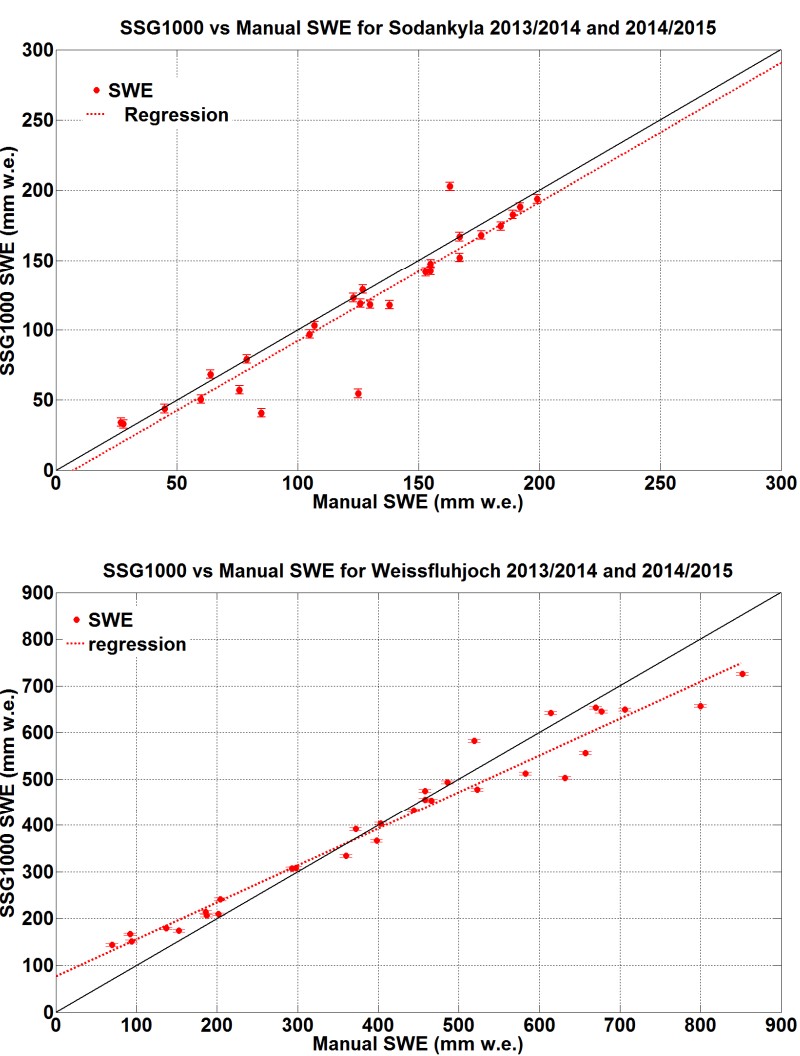

Figure5: SSG1000 vs Manual SWE at Sodankylä (top) and Weissfluhjoch (bottom) for the 2013/2014 and 2014/2015 seasons. Black line is 1:1. Error bars represent manufacturer's stated sensor accuracy.

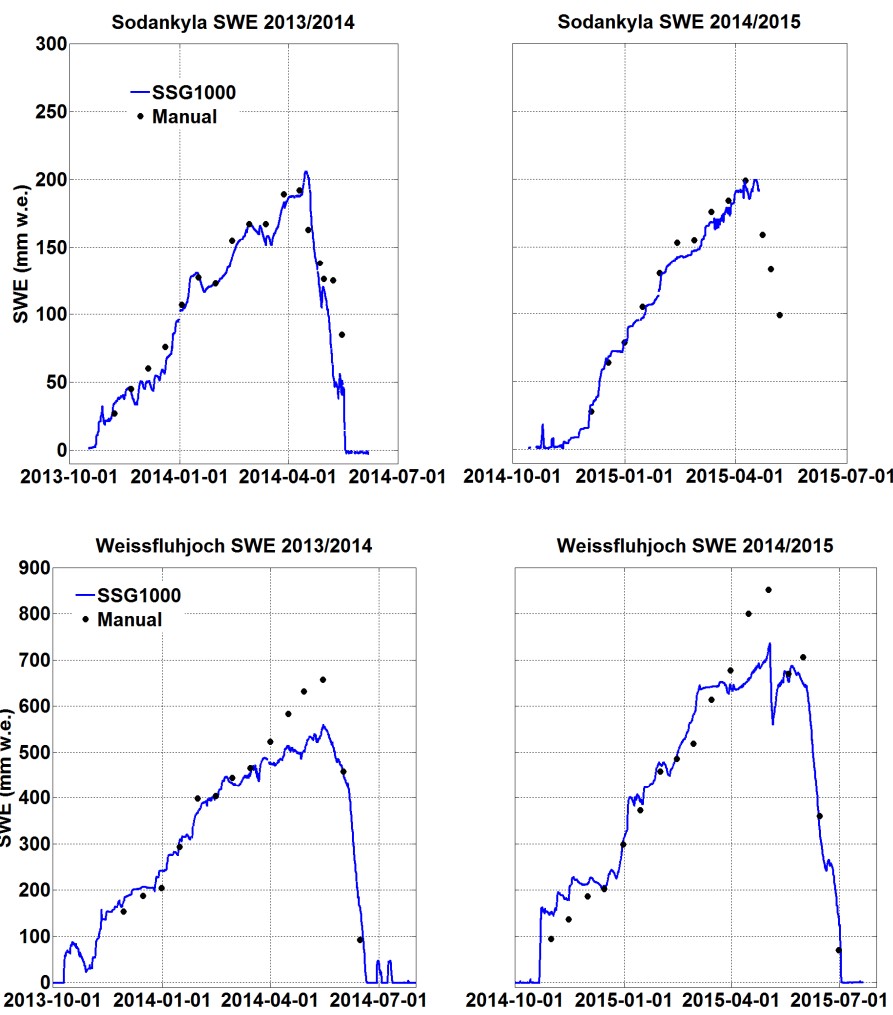

Figure 8: Time series of the SSG1000 SWE sensors and manual SWE measurements at Sodankylä (top) and Weissfluhjoch (bottom) for the 2013/2014 (left) and 2014/2015 (right) seasons.

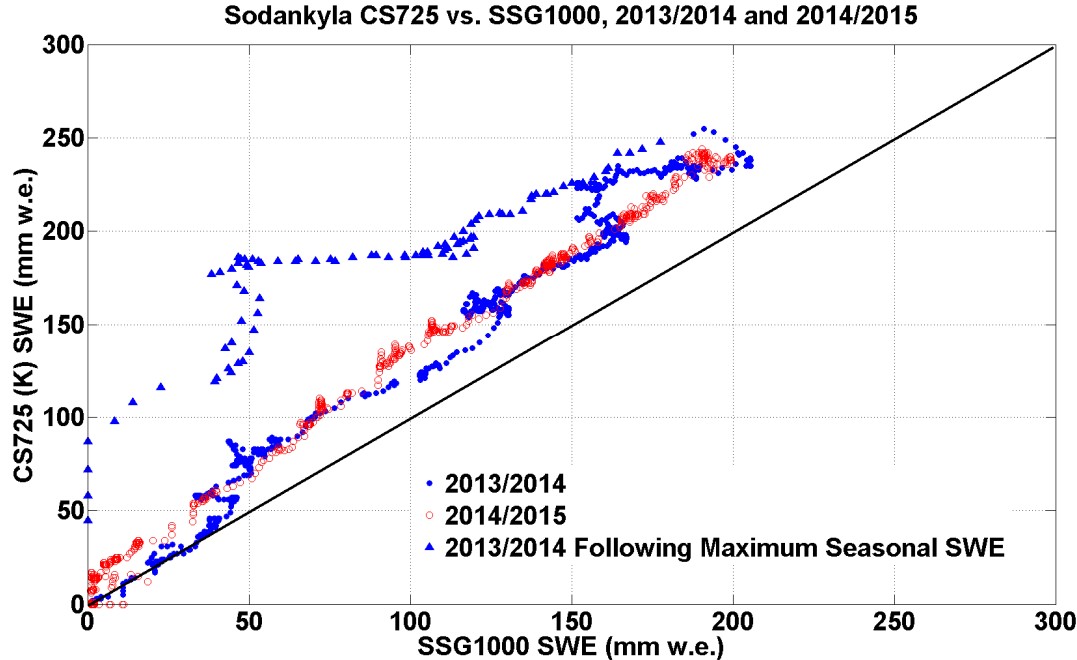

Figure 9: CS725 vs SSG1000 for the 2013/2014 (blue dots/blue triangles) and 2014/2015 (red) seasons at
Sodankylä. Black line is 1:1.