# Peer review of "An assessment of two automated snow water"

_The Cryosphere, 2016_

## Referee Comment (RC1) · Anonymous Referee #1 · 20 May 2016

Although snow water equivalent (SWE) is very important information for not only disaster forecasting but also for earth science, there is still room for discussion of the methods of automatic SWE measurement. They compared different automatic SWE measurement methods (CS725, SSG1000), which are based on the different principles, with the manual SWE measurement at three sites. Then they discussed the characteristics of the comparison results at each site. Although I am very interested in their works and do not doubt that their works give the basic important information for the improvement of automatic SWE measurement methods, their work only shows few scientific new findings in the present version. From this view, I think their topic

should be suitable for rather GI (or some publications which mainly treat the topics of instrument, method) publication than TC publication. In order for this manuscript to be accepted for TC, the authors should completely reconstruct the manuscript to clear the new scientific findings and scientific contribution of their works.
* * *

---

## Referee Comment (RC2) · Anonymous Referee #2 · 6 Jun 2016

The paper compares SWE manual measurements with automated sensors of attenuation of passive gamma radiation attenuation device in two SPICE sites and another site with rather different climatic and snow conditions; and also one scale snow in one of the SPICE sites. The topic is of high interest for a broad community as the sensors that are compared are starting to be very popular in many sites of the world, and it is necessary to discuss about their accuracy, possible sources of uncertainty, etc. Due to the limited length of snow observations, few locations analysed and some problems related with the experimental design it is not possible to give strong evidences on their accuracy and the reason of biases found between manual and automated measure-

ments. However, I think that the paper still has enough interest for many readers, and it launches some hypothesis of interest that may serve as basis for further research.

In my opinion, the structure of the manuscript is not the best to present the results. I recommend to reorganize the presentation of them showing the equivalent figures for each site together, instead of doing subsections of the results for each site (indeed the discussion is presented in the way I suggest). Meanwhile table 1 and 2 can be combined (and added results from Fortress). In this way it would be reduced the final number of figures (the 14 current figures is excessive in my opinion). More important, it would be possible to identify common processes amongst sites and their differences, and the paper would gain consistency (in the current version some figures are made for one site but not for the others..e.g.). Thus it could be presented the validation of CS725 and SSG1000 (where available) in the three sites with a couple of multi-panel plots (one panel per site), and afterwards to show figures that allow explaining the patterns of accuracy/error shown (the figures relating air temperature and difference SWE, the soil moisture...

- One concern is how to ensure that snow depth in automated sensors is the same than that were SWE is measured manually in the three presented sites. This could be another source of error not mentioned in the manuscript. In page 11 is mentioned the spatial variability in snow melting that could affect to different snow depth. Are there snow depth sensors installed above the measured areas with automated sensors. If this is the case, we could see how well the depths are similar and if there are differences, some plot could focus on estimated snow density.

-Is there any evidence of a relationship between snow depth or amount of SWE with bias with the manual measurements. There are some references that CS725 may be inaccurate under very deep snowpack. - I think that Figure 11 should show the bias between manual and automated measurements to properly observe the coincidence between liquid water content and SWE differences.

- It seems that the existence of water around SSG1000 may cause serious disruptions in the functioning of the device. Is it apparently due to problems in their installation or is a problem of design of the device?

Hoping my comments will result useful.
* * *

---

## Referee Comment (RC3) · C. Fierz (Referee) · 7 Jul 2016

**Review**   for 'The Cryosphere', Jul 2016.

Paper tc-2016-57, "*An assessment of two automated snow water equivalent instruments during the WMO Solid Precipitation Intercomparison Experiment*", by Smith et al..

**Comments to the editor**

**General comments**

The goal of this paper, as stated by the authors, is, *"to assess the use and accuracy of two instruments that were tested during the WMO-Solid Precipitation Intercomparison Experiment the Campbell Scientific CS725 and the Sommer Messtechnik SSG1000 snow scale"* as well as, "*to inform users of the best way to use these instruments and of any potential measurement issues that may influence their data interpretation.*" Unfortuately, however, I don't feel I get the promised info by reading that paper. The deficiencies of the CS725 (first 5 lines on page 4) are simply confirmed and no convincing, in depth analysis of possible source of errors are addressed for any of the two instruments. Instead, spatial variability is invoked to explain the mismatch between the continuous measurements and the manual, punctual (in time) reference measurements, the error of which are not quantified either.

In view of the above and my comments below, I can hardly recommend to accept that paper for publication. Indeed, I really doubt that the authors have enough convincing arguments and data to bring the paper in line with their goals, even after major revisions.

**Specific comments**

- p. 1, lines 25-26:   *"These manual measurements are considered to be the reference for the intercomparison."* This is one of the crux of that paper. The devices are hardly looked at while a whole lot of blame goes on these manual measurements, the error of which are hardly addressed though.
- p. 1, line 30:   *"throughout the intercomparison periods"* is absolutely misleading and false. One full ablation period is missing and the problems of the instruments were not looked at.
- p. 1, line 33:   *"seasonal melt"* suggest replacing by "**ablation period**" [throughout the paper]. Furthermore, is pre-melt = accumulation period? I strongly suggest that you define these terms properly once and use them consistently throughout the paper. See for example on p. 6, line 31 for "*point of maximum seasonal SWE*"
- p. 3, line 13:   "*2 Instrumentation and Methods*" Should try to not give interpretation in that paragraph but include it in the results section, for example as "previous intercomparison"
- p. 3-*4*; lines 26-*5* :   Is this the correct place for such comments? Should be moved to discussion part as an introduction to it.
- p. 4, line 15:   "*… impact of the move are considered to be negligible.*" Why? Later you speak of spatial variability influencing the results.
- p. 4, lines 29-30:   "*... to stabilize the overlying snowpack and prevent ice bridging.*" Why does the snowpack need to be stabilized? How is ice bridging prevented? What observations do corroborate this?
- p. 5, line 3:   "*..., and the only snow scale provided ...*" is incorrect. There is another SPICE site (Weissfluhjoch) equipped with a snow scale … and a snow pillow next to it from the same provider.
- p. 5, line 8:   "*... reliable manner ...*" but not always. The simple regression does not reveal the true problems!
- p. 5, lines 13-15 :   "The sensitivity …" Such a sentence belongs to the summary and conclusion section.
- p. 5, lines 24-25:   "*... has a mean measurement error less than 0.5 %.*" 0.5 % of what? Does this refer to the repeatability of measurements? Overall, the number looks very optimistic and the reference Farnes 1983 is hardly available. From other publications by the same author (1980 and 1982, see Kinar & Pomeroy, 2015b), this figure can hardly be reproduced. I'd strongly suggest to be more precise here.
- p. 5, line 25:   "*were taken just inside the footprint of the CS725*" How do these disturbances affect the measurements?

- p. 7, lines 14-15 :    "*the instrument trends are the same as for the manual measurements*" In my view your simple regression analysis fails here and does not look at problematic features. For example, how do you explain the apparent loss of mass around mid February 2014? Similar unexpected wiggles are also seen at other times on both seasons. These spurious measurements are also known to occur at Weissfluhjoch and are not to be expected from a well designed, continuously recording device.

- p. 7, lines 15-21:    This comparison or 'tracker' does not appear to work very well. Indeed, in January 2014 there is a large increase in the 'Difference in SWE' while air temperature plummets! Similar behaviour can be found at other times. In summary, there is another reason behind these large increases, but which?

- p. 11, line 11:    "*systematic sampling errors*" Can these be avoided?

- p. 11, lines 24-27:    "*Although ...*" A somehow simplistic view. In the paper you never assess any of the errors you assign the outliers to. This is definitely the biggest weakness of that paper.

- p. 11, lines 30-36:    Poor conclusions! What does this linear relationship show? Would you calibrate the CS725 with a SSG1000? Were the deficiencies of the CS725 not already known (see your introduction)?

- p. 12, line 4:    "*... not all increases in the bias ...*" Interesting, you don't even mention those in the discussion!

- p. 12, line 8:    "*... errors in the manual SWE measurements. ...*" I agree that measurement errors can amount to a certain percentage of SWE. But you don't even quantify these errors, even though you use them as reference. Blaming not quantified errors for the observed mismatch seems simplistic indeed.

- p. 12, line 32:    "*... have a good agreement ...*" Here I have really hard times to follow the logic of your conclusions. First you blame the manual measurements for mismatch and then you claim a good agreement!

**Minor comments**

- Please ALWAYS put a space between numbers and units (often wrong)
- p. 1; line 28:    Replace "*(w.e.)*" by "(mm w.e.)".
- p. 1; line 29:    Replace "*Creek respectively*" by "Creek, respectively".
- p. 2; line 7:    Are these two units equivalent?
- p. 3; line 1:    Replace "*(SPICE; Nitu*" by "SPICE (Nitu".
- p. 3; line 28:    Replace "*snow cores*" by "snow courses".
- p. 3; lines 28-29:    What are "*snow pit densities*"? Please describe. Sampler?
- p. 4; line 30:    "*0.3 % of full scale*" that is 3 mm w.e.! Under what conditions? Moreover, the high resolution seems useless.
- p. 4, line 35:    What is drifting? Snow? The electronics?
- p. 5; line 23:    "*snow tube*" Is this the correct term? I'd suggest using "snow sampler" - as found elsewhere in the literature – throughout your paper.
- p. 7; line 5:    Replace "*almost*" by "by almost".
- p. 7; line 29:    Replace "*offset*" by "intercept", throughout, as used in the tables.
- p. 7; line 32:    What does "*differential*" mean?
- p. 8; line 8:    Try to read "*2013/2014 only due to data unavailability for 2014/2015*" and replace that sentence.
- p. 8; line 30:    "*melt and re-freeze occurred*" That really questions the term "*pre-melt*" used elsewhere in the paper.
- p. 9; lines 32-33 :    "*gravimetric water fraction*" vs "*volumetric water content*" What is the relation?

- p. 10; line 27:     "*bias drop*" Do you mean the -39 kg m$^{-2}$ value? Hard to tell whether the short, hardly above freezing period or the following sub-freezing period is responsible for that, isn't it?
- p. ; line :     Replace "" by "".
- p. ; line :     Replace "" by "".
- p. ; line :     Replace "" by "".

**Tables**

Tables:          Replace "*multi-season*" by "two seasons"
Tables:          "*Combined*" What is combined?

**Figures**

Figures :        *To avoid confusion, use an ISO date format, for example, 2014-04-01 for 1 Apr 2014. Is "UTC" the unit of date? As no hour time stamps are shown, you may drop it altogether.*
Figure 5:        The division by two of the "Difference in SWE" is somewhat annoying. You could also use the right-hand axis to avoid this.

2016-07-07 / Charles Fierz

---

## Author Comment (AC1) · 19 Aug 2016

**An assessment of two automated snow water equivalent instruments during the WMO Solid Precipitation Intercomparison Experiment**

Craig D. Smith, Anna Kontu, Richard Laffin, and John W. Pomeroy

We would like to thank the referees for taking the time to review this paper submission. Your comments and suggestions are most appreciated and we have tried to address all of them in turn, providing clarification and revisions where necessary. Our responses are highlighted in the text below. Incorporating your suggestions and addressing your comments and concerns have certainly helped to improve this manuscript.

**Response to referee comments**

**Anonymous Referee #1**

Although snow water equivalent (SWE) is very important information for not only disaster forecasting but also for earth science, there is still room for discussion of the methods of automatic SWE measurement. They compared different automatic SWE measurement methods (CS725, SSG1000), which are based on the different principles, with the manual SWE measurement at three sites. Then they discussed the characteristics of the comparison results at each site. Although I am very interested in their works and do not doubt that their works give the basic important information for the improvement of automatic SWE measurement methods, their work only shows few scientific new findings in the present version. From this view, I think their topic should be suitable for rather GI (or some publications which mainly treat the topics of instrument, method) publication than TC publication. In order for this manuscript to be accepted for TC, the authors should completely reconstruct the manuscript to clear the new scientific findings and scientific contribution of their works.

**Author's response**

We will reconstruct the Results section and arrange the text by instrument, not by site, as suggested by Referee #2. We hope that this change will highlight our results including the new findings that came out of the SPICE intercomparison.

**Author's changes in manuscript**

- Text in **3 Results and Discussion** arranged by instrument and divided into two sections, **3 Results** and **4 Discussion**.

**Anonymous Referee #2**

The paper compares SWE manual measurements with automated sensors of attenuation of passive gamma radiation attenuation device in two SPICE sites and another site with rather different climatic and snow conditions; and also one scale snow in one of the SPICE sites. The topic is of high interest for a broad community as the sensors that are compared are starting to be very popular in many sites of the world, and it is necessary to discuss about their accuracy, possible sources of uncertainty, etc. Due to the limited length of snow observations, few locations analysed and some problems related with the experimental design it is not possible to give strong evidences on their accuracy and the reason of biases found between manual and automated measurements. However, I think that the paper still has enough interest for many readers, and it launches some hypothesis of interest that may serve as basis for further research.

In my opinion, the structure of the manuscript is not the best to present the results. I recommend to reorganize the presentation of them showing the equivalent figures for each site together, instead of doing subsections of the results for each site (indeed the discussion is presented in the way I suggest). Meanwhile table 1 and 2 can be combined (and added results from Fortress). In this way it would be reduced the final number of figures (the 14 current figures is excessive in my opinion). More important, it would be possible to identify common processes amongst sites and their differences, and the paper would gain consistency (in the current version some figures are made for one site but not for the others..e.g.). Thus it could be presented the validation of CS725 and SSG1000 (where available) in the three sites with a couple of multi-panel lots (one panel per site), and afterwards to show figures that allow explaining the patterns of accuracy/error shown (the figures relating air temperature and difference SWE, the soil moisture...

One concern is how to ensure that snow depth in automated sensors is the same than that were SWE is measured manually in the three presented sites. This could be another source of error not mentioned in the manuscript. In page 11 is mentioned the spatial variability in snow melting that could affect to different snow depth. Are there snow depth sensors installed above the measured areas with automated sensors. If this is the case, we could see how well the depths are similar and if there are differences, some plot could focus on estimated snow density.

Is there any evidence of a relationship between snow depth or amount of SWE with bias with the manual measurements. There are some references that CS725 may be inaccurate under very deep snowpack. - I think that Figure 11 should show the bias between manual and automated measurements to properly observe the coincidence between liquid water content and SWE differences.

It seems that the existence of water around SSG1000 may cause serious disruptions in the functioning of the device. Is it apparently due to problems in their installation or is a problem of design of the device?

Hoping my comments will result useful.

Author's response

The referee points out the limited length of observations, few locations, and problems with experimental design. This was largely related to the choice of limiting analysis to the SPICE instrumentation over the SPICE intercomparison period and using available reference measurements. We did add CS725 data from Fortress Mountain (which is not a SPICE site) to help support our theory that CS725 offsets were due to changes in soil moisture related to sandy soil substrates. To have more data and locations, we will add one more site, Weissfluhjoch, to the SSG1000 intercomparison which adds 2 seasons of instrument intercomparisons. However, as the referee points out in regards to the existing intercomparison, the methods for the manual reference measurements differ from the other sites. This contributes to the difficulty in assessing the level of uncertainty in the reference.

The referee suggests looking at automated snow depths if they are measured in the same field of view as the SWE measurements. This, unfortunately, was not the case as the snow depth sensors were located at different locations participating in their own assessment. This deficiency is noted and we will comment that this could assist future SWE intercomparisons. We did use manual snow depth measurements at Caribou Creek and Sodankyla to estimate site variability and will add a commentary on this.

As suggested, we will rearrange the Results and Discussion section by instrument, not by site. We will also combine relevant figures and tables to reduce their number and to make comparisons between sites easier.

The problems with SSG1000 and water are due to a design feature: the cables are so short that the electronics box must be installed below the instrument on the ground. We know that Fortress Mountain site has also experienced similar problems with the instrument. After SPICE, the Sodankylä team asked the manufacturer to replace the cables with longer ones. A third measurement winter was without instrument failures, as the electronics box was now mounted about 50 cm above ground. This will be briefly mentioned in the text.

We revised Figure 11 (now Figure 6) to include the difference between the CS725 and manual SWE measurements. This now shows the relative timing between the changes in the soil moisture/soil temperature and the changes in the instrument offset. However, the difference in the frequency of measurements means that interpretation isn't necessarily clear. As the manuscript states, it offers an explanation for some of the initial offset which is shown by the first intercomparison in mid-December but the large discrepancies shown in mid-February are not attributed to changes in liquid water in the soil anyways. It does mark the coincidence between spring soil thaw and infiltration of liquid water and a corresponding increase in the sensor bias.

**Author's changes in manuscript**

- Text in **3 Results and Discussion** arranged by instrument and relevant figures and tables combined. Text divided into two sections, **3 Results** and **4 Discussion**.
- Weissfluhjoch is added to the SSG1000 intercomparison.
- The SSG1000 problems will be addressed in the text.
- Revised Figure 11 (now Figure 6) to include a plot of the difference in SWE between the CS725 and the manual measurements in relation to the soil moisture/soil temperature changes
- We added further commentary in the Discussion section on how the experiment design could be improved for future SWE intercomparisons, including measuring snow depth in the same footprint as the SWE sensors.
- Added a commentary in the Discussion section on the snow depth variability at Caribou Creek and Sodankyla as an indicator of how the SWE could also vary with space and time.

**Referee #3 Charles Fierz**

**General comments**

The goal of this paper, as stated by the authors, is, "to assess the use and accuracy of two instruments that were tested during the WMO - Solid Precipitation Intercomparison Experiment the Campbell Scientific CS725 and the Sommer Messtechnik SSG1000 snow scale" as well as, "to inform users of the best way to use these instruments and of any potential measurement issues that may influence their data interpretation." Unfortuately, however, I don't feel I get the promised info by reading that paper. The deficiencies of the CS725 (first 5 lines on page 4) are simply confirmed and no convincing, in depth analysis of possible source of errors are addressed for any of the two instruments. Instead, spatial variability is invoked to explain the mismatch between the continuous measurements and the manual, punctual (in time) reference measurements, the error of which are not quantified either. In view of the above and my comments below, I can hardly recommend to accept that paper for publication.

Indeed, I really doubt that the authors have enough convincing arguments and data to bring the paper in line with their goals, even after major revisions.

**Author's response**

It is true that we confirm some deficiencies in the CS725, as the importance for soil moisture calibration of the CS725 is previously established (Martin et al. 2008; Wright et al. 2011). However, in previous comparisons the instrument has agreed much better with other measurements (automated or manual) than we have observed during SPICE. Our analysis shows that the CS725 assessed during SPICE did not compare as well with manual measurements as did previous intercomparisons and we propose that the larger offset is due to changes in soil moisture content after calibration and throughout the snow season. This was in turn caused by the soil type at both intercomparison sites which was predominately sand. The possible ways for soil moisture to change during winter are identified by Gray et al. (1985, 2001) and Lilbaek and Pomeroy (2008). We argue that these soil conditions did not exist for previous intercomparisons and instrument users need to be made aware of this. We also imply that this is not necessarily an instrument deficiency and some hydrological users may be able to use this measurement principle to their advantage.

We agree with the need of more in-depth analysis to support the hypotheses, and will provide this in a revised manuscript or make recommendations to guide future SWE sensor intercomparisons to fill these deficiencies.

The specific and minor comments are addressed after each comment. The resulting changes to the manuscript are also addressed after each comment.

**Specific comments**

•p. 1, lines 25-26: "These manual measurements are considered to be the reference for the intercomparison." This is one of the crux of that paper. The devices are hardly looked at while a whole lot of blame goes on these manual measurements, the error of which are hardly addressed though.

We now address some of the error in the manual measurements in the Discussion section. The literature addresses the mean bias associated with some manual samplers but the variability of the potential error in field sampling is quite large and depends highly on the skill of the user and the condition of the snow pack.

•p. 1, line 30: "throughout the intercomparison periods" is absolutely misleading and false. One full ablation period is missing and the problems of the instruments were not looked at.

One ablation period is missing for one sensor only…the SSG1000 at Sodankyla for the 2014/2015 period. However, we recognize that this limits the sample size of the intercomparison during ablation for this sensor (and for intercomparison with the other sensor). We will change the text "throughout the intercomparison periods" to "when data were available" where applicable. In addition, we will add SSG1000 intercomparison from an additional site

•p. 1, line 33: "seasonal melt" suggest replacing by "ablation period" [throughout the paper]. Furthermore, is pre-melt = accumulation period? I strongly suggest that you define these terms properly once and use them consistently throughout the paper. See for example on p. 6, line 31 for "point of maximum seasonal SWE"

We agree that consistent terminology is important. We will replace 'pre-melt" with "accumulation period" and "seasonal melt" with "ablation period".

•p. 3, line 13: "2 Instrumentation and Methods" Should try to not give interpretation in that paragraph but include it in the results section, for example as "previous intercomparison"

The commentary about previous intercomparisons has been moved to the "Introduction" section and all interpretation has been moved to "Results" or "Discussion".

•p. 3-4; lines 26-5: Is this the correct place for such comments? Should be moved to discussion part as an introduction to it.

Agreed

•p. 4, line 15: "... impact of the move are considered to be negligible." Why? Later you speak of spatial variability influencing the results.

We resolve this by including commentary on the spatial variability at this site and demonstrate how small this is over the distances of the sensor move.

•p. 4, lines 29-30: "... to stabilize the overlying snowpack and prevent ice bridging." Why does the snowpack need to be stabilized? How is ice bridging prevented? What observations do corroborate this?

This is a statement made by the manufacturer (and is stated as such) to justify their design of a larger platform with a smaller platform in the centre that does the actual weight measurement. It is out of our scope to validate their design other than to comment on the quality of the measurements.

•p. 5, line 3: "..., and the only snow scale provided ..." is incorrect. There is another SPICE site (Weissfluhjoch) equipped with a snow scale ... and a snow pillow next to it from the same provider.

The text in the manuscript is correct. The snow scale in Weissfluhjoch was provided by the site, not by manufacturer for testing during SPICE. However, based on this comment as well as some other comments about lack of snow scale intercomparison data (due to instrument failure), we have chosen to add the snow scale intercomparison for Weissfluhjoch..

•p.5, line 8: "... reliable manner ..." but not always. The simple regression does not reveal the true problems!

This statement is in reference to the actual functionality of the instrument (see the context) rather than an assessment of the measurement accuracy. The instrument functioned with a low failure rate.

•p. 5, lines 13-15: "The sensitivity..." Such a sentence belongs to the summary and conclusion section.

We agree. The sentence will be moved.

•p. 5, lines 24-25: "... has a mean measurement error less than 0.5 %." 0.5 % of what? Does this refer to the repeatability of measurements? Overall, the number looks very optimistic and the reference Farnes 1983 is hardly available. From other publications by the same author (1980 and 1982, see Kinar & Pomeroy, 2015b), this figure can hardly be reproduced. I'd strongly suggest to be more precise here.

Farnes et al. (1982) state that ESC-30 overmeasures by -0.3 % (i.e. undermeasures) of the true SWE, and that the correction factor for ESC-30 is 1.00 (no correction required). The accuracy is quoted by Goodison et al. (1987) which was added as a reference. This paper may be more available than the original Farnes et al.

paper. Of course, these errors are in ideal situations, as stated by Kinar & Pomeroy who cite Powell (1987) in reference to errors in measuring more difficult snow packs. We add a discussion on this as it relates to measurements during SPICE (especially at Caribou Creek).

•p. 5, line 25: "were taken just inside the footprint of the CS725" How do these disturbances affect the measurements?

Since the sample is 30 cm$^2$ inside an 80 m$^2$ sensor footprint, the impact is negligible but the sample area was filled in with discarded snow when possible. This was clarified in the text.

•p. 7, lines 14-15: "the instrument trends are the same as for the manual measurements" In my view your simple regression analysis fails here and does not look at problematic features. For example, how do you explain the apparent loss of mass around mid February 2014? Similar unexpected wiggles are also seen at other times on both seasons. These spurious measurements are also known to occur at Weissfluhjoch and are not to be expected from a well designed, continuously recording device.

Mid feb 2014, March 2014, March-April 2015: very cold periods (-30 C) after positive air temperatures resulted in ice bridging. The snow supports itself and the weight is not on the load cell. After adding the Weissfluhjoch data, we see it here as well (as the reviewer points out). This issue will be discussed in the revised paper in the Discussion section.

•p. 7, lines 15-21: This comparison or 'tracker' does not appear to work very well. Indeed, in January 2014 there is a large increase in the 'Difference in SWE' while air temperature plummets! Similar behaviour can be found at other times. In summary, there is another reason behind these large increases, but which?

Agreed, the correlation between temperature increase and the immediate corresponding increase in sensor bias isn't always clear. Sources of error are now presented in the Discussion section, including the potential issues with manually sampling a snow pack that persists after a melting period as a result of ice layers, etc,

•p. 11, line 11: "systematic sampling errors" Can these be avoided?

I see the reviewer's point. This was poorly worded. These sampling errors could be systematic but not necessarily so. The text was changed.

•p. 11, lines 24-27: "Although ..." A somehow simplistic view. In the paper you never assess any of the errors you assign the outliers to. This is definitely the biggest weakness of that paper.

The text referenced here has been changed and we now address the errors in more detail, including ice bridging, in the Discussions section.

•p. 11, lines 30-36: Poor conclusions! What does this linear relationship show? Would you calibrate the CS725 with a SSG1000? Were the deficiencies of the CS725 not already known (see your introduction)?

The linear relationship is meant to show that it's not just the manual measurements causing the bias but rather the measurement principle of the sensor. We have tried to make this clearer in both the Discussion and the Conclusion sections. The previous literature does point out some known deficiencies in the CS725 but the behavior seen at Sodankyla, especially during melt, has never been documented. The comparison with the SSG1000 supports our conclusions that the increased bias is related to infiltration of meltwater into the sandy soils, and serves as a warning to users who may be using the instrument in similar situations.

•p. 12, line 4: "... not all increases in the bias... " Interesting, you don't even mention those in the discussion!

==This has been corrected, see note above about sources of error==

•p. 12, line 8: "... errors in the manual SWE measurements...." I agree that measurement errors can amount to a certain percentage of SWE. But you don't even quantify these errors, even though you use them as reference. Blaming not quantified errors for the observed mismatch seems simplistic indeed.

==It is difficult to estimate the measurement error in the field because it highly depends on the measurement conditions and the observer, however we do attach some loose estimates on this error in the Discussions section on Manual SWE measurements.==

•p. 12, line 32: "... have a good agreement ..." Here I have really hard times to follow the logic of your conclusions. First you blame the manual measurements for mismatch and then you claim a good agreement!

==The point was that there was a good agreement considering the potential issues with the manual measurements and the measurement principle of the sensor. Hopefully the revised text makes this clearer.==

**Minor comments**

•Please ALWAYS put a space between numbers and units (often wrong)

==We will correct these.==

•p. 1; line 28: Replace "(w.e.)" by "(mm w.e.)".

==We will replace this.==

•p. 1; line 29: Replace "Creek respectively" by "Creek, respectively".

==We will correct this.==

•p. 2; line 7: Are these two units equivalent?

==Yes they are, assuming that density of water is 1 kg/m3.==

•p. 3; line 1: Replace "(SPICE; Nitu" by "SPICE (Nitu".

==Replaced with "(SPICE) (Nitu"==

•p. 3; line 28: Replace "snow cores" by "snow courses".

==No, they actually compared to 8 snow samples (cores), not 8 snow courses.==

•p. 3; lines 28-29: What are "snow pit densities"? Please describe. Sampler?

==It means that a density profile of 5 cm high samples is measured, SWE is determined for each sample and the total SWE is calculated as a sum of the layers. This is thought to be more accurate than SWE from bulk snow sample (core). Details will be added to the revised manuscript.==

•p. 4; line 30: "0.3 % of full scale" that is 3 mm w.e.! Under what conditions? Moreover, the high resolution seems useless.

This is what the manufacturer tells about the technical details of their instrument. They do not specify the conditions, or give reasons for such high resolution.

•p. 4, line 35: What is drifting? Snow? The electronics?

This was in reference to drifting snow. This has been clarified in the text.

•p. 5; line 23: "snow tube" Is this the correct term? I'd suggest using "snow sampler" -as found elsewhere in the literature –throughout your paper.

Snow sampler can be any kind of sampler, from 5 cm high wedge sampler to a long tube. "Snow tube" defines the type of sampler used, and is a widely used term (e.g. by Kinar and Pomeroy 2015).

•p. 7; line 5: Replace "almost" by "by almost".

We will replace this.

•p. 7; line 29: Replace "offset" by "intercept", throughout, as used in the tables.

We will replace this.

•p. 7; line 32: What does "differential" mean?

Different melt rates at different locations at the site.

•p. 8; line 8: Try to read "2013/2014 only due to data unavailability for 2014/2015" and replace that sentence.

Clarified this sentence

p. 8; line 30: "melt and re-freeze occurred" That really questions the term "pre-melt" used elsewhere in the paper.

'Onset of snow-melt' and 'snow-melt season' are commonly used terms for the final snow melt (ablation) in the spring. They do not rule out the possibility of short melt-refreeze cycles during the accumulation period. However, we will replace 'pre-melt' with 'accumulation period' in the text for clarity.

•p. 9; lines 32-33: "gravimetric water fraction" vs "volumetric water content "What is the relation?

Volumetric water content (theta) is the volume of water divided by the total volume. Gravimetric water content or fraction (u) is the mass of water divided by dry soil mass. They are related by theta = u SG, where SG is the soil specific gravity and depends on its density.

**Author's changes in manuscript**

Changes to the manuscript are integrated into the specific responses to the referee's comments above.

---

## Referee Report (RR1)

[referee-annotated manuscript omitted]

---

## Author Response (AR2)

**Editor Decision: Reconsider after major revisions** (23 Aug 2016) by Dr. Samuel Morin Comments to the Author: Dear authors,

Thank you for submitting a revised version of the article along with detailed replies to the authors comments. I will send the ms for further review to the reviewers, but before this I'm asking you to perform a few changes to the manuscript, which I hope will improve it and make the further review smoother.

Here are the changes requested:

- The new structure of the manuscript seems more appropriate and corresponds to the recommendations of the reviewers. However I'm asking you to remove from Section 2. any material which corresponds to "results", "conclusions" or "implications" of the intercomparison. This applies in particular to the following statement :

\* Page 6, line 19 : "The instruments performed in a reliable manner exhibiting a measurement rate higher than 95 % at both sites over the course of the two winter seasons. No malfunctions were noted and no maintenance was required." This is a result, not an instrument description.

**We made this change by creating Section 4.2 "Sensor Reliability" in the Discussion section of the manuscript where we report on the failure rate of the sensors**

\* Page 8, line 4 : "The instruments at both sites worked in a reliable manner during the accumulation periods but the instrument at Sodankylä malfunctioned due to water damage to the electronics late in the spring of 2014 and again early spring of 2015. At Weissfluhjoch, 99% of the 1-minute data for both years are usable for intercomparison while 83 % and 67 % of the 1-minute data are usable for the concurrent intercomparison periods at Sodankylä. Other than this, no malfunctions were reported or maintenance required during the intercomparison." This is a result, not an instrument description.

**<mark>See above</mark>**

\* Page 8, line 10 : "The electronics box of the SSG1000 is designed to be installed below the instrument on the ground, which could be flooded during snowmelt. After SPICE, the manufacturer was asked to provide longer cables allowing the installment of the electronics box about 0.5 m above the ground. After this modification, there were no problems with water and electronics during the following snowmelt season." These statements mix instrument setup description with results and recommendations. Better wording and location of the information should be found ; in addition the data for the season 2015/2016 are not included in the analysis so the statement about the fix s not substantiated by the data analysis. Please consider adding the corresponding data to the ms, or place this information towards the end of the ms (conclusions or implications).

See above. The comment about the sensor re-design was added to Section 4.2. We also removed the comment about the following snow season. Instead, we added the sentence "This modification should

help prevent water damage to the electronics and therefore resolve the issue."

- There are a few structure issues in the Results section, in particular the fact that there is a section 3.1.2.1 without a 3.1.2.2 section which means that the section 3.1.2.1 could be simply a paragraph, or that section 3.1.2 could be split in more subsections.

**We removed the sub-section heading for 3.1.2.1 and it now simply becomes 3.1.2.**

- Last, I'm asking you to enlarge significantly all font sizes in the Figures, taking into account that most Figures in TC articles are csingle column width, and should still be readable.

We increased the font size for the figures and we hope that this works better with the final layout. Of course, we can always further modify the figure fonts and layout before final submission if necessary.

I thank you for performing the changes, the then-revised manuscript will be sent for review to the same reviewers as the previous round.

**An assessment of two automated snow water 2 equivalent instruments during the WMO Solid**

- 3 Precipitation Intercomparison Experiment
- 4

5 Craig D. Smith1, Anna Kontu2, Richard Laffin3, John W. Pomeroy4

6 1Environment Canada, Saskatoon, S7N 3H5, Canada2FinnishCanada

7 2Finnish Meteorological Institute, Sodankylä, FI-99600, Finland

- 8 3Campbell Scientific, Edmonton, T5L 4X4, Canada

[revised manuscript text omitted]

|           |                             |              | ε             | -          | RMSE       | Mean     |          |                     |                                          |
|-----------|-----------------------------|--------------|---------------|------------|------------|----------|----------|---------------------|------------------------------------------|
| Sito      | Season                      | β            | к(ті)         | r²         | к(ті)      | Rolativo | •
n • | {u                  | Formatted: Line spacing: 1.5 lines       |
| Site      | 563501                      | К(ТІ)        | mm w.e.       | K(TI)      | mm w.e. E  | Bias (K) |          |                     | Formatted: Left, Line spacing: 1.5 lines |
| Sodankylä | 2013/2014                   | 1.24(1.27)   | 8.77(3.17)    | 0.92(0.92) | 43.0(42.2) | 30.1%    | 17 🔸     | [                   | Formatted: Line spacing: 1.5 lines       |
|           | 2013/2014
(accumulation) | 1.24(1.28)   | 0.0123(-6.63) | 0.97(0.97) | 35.6(33.9) | 24.6%    | 13 •     |                     | Formatted: Line spacing: 1.5 lines       |
|           | (accumulation)              |              |               |            |            |          |          | کي ل         | Formatted: Line spacing: single          |
|           | 2014/2015                   | 1.06(1.13)   | 26.9(24.2)    | 0.96(0.96) | 36.6(42.2) | 30.9%    | 13 🔸     | `\_` ` [     | Formatted: Line spacing: 1.5 lines       |
|           | 2014/2015                   | 1.05(1.12)   | 23.3(20.2)    | 0.99(0.99) | 30.0(35.7) | 28.1%    | 10 🔸     | ، ِ ` ا      | Formatted: Line spacing: 1.5 lines       |
|           | (accumulation)              |              |               |            |            |          | 4        |                     | Formatted: Line spacing: 1.5 lines       |
|           | Combined                    | 1.16(1.21)   | 16.8(11.9)    | 0.92(0.92) | 40.3(42.2) | 30.4%    | 30 🔸     |                     | Formatted: Line spacing: single          |
| Caribou   | 2013/2014                   | 0.783(0.764) | 40.6(46.9)    | 0.78(0.72) | 22.8(27.5) | 22.2%    | 12 🔸     |                     | Formatted: Line spacing: 1.5 lines       |
| Creek     | 2013/2014                   | 0.982(0.997) | 17.7(20.2)    | 0.79(0.75) | 18.0(22.2) | 15.4%    | 9 🔸      | ``` [        | Formatted: Line spacing: 1.5 lines       |
|           | (accumulation)              |              |               |            |            |          | +        |                     | Formatted: Line spacing: 1.5 lines       |
|           | 2014/2015                   | 0.849(0.849) | 27.1(30.4)    | 0.77(0.71) | 23.6(27.4) | 63.0%    | 7 🔸      |                     | Formatted: Line spacing: 1.5 lines       |
|           | 2014/2015                   | 1.12(1.31)   | -8.38(-14.5)  | 0.55(0.60) | 25.4(29.5) | 42.4%    | 4 🔸      | ```` [       | Formatted: Line spacing: single          |
|           | (accumulation)              |              |               | . ,        | . ,        |          | 4        | ), ` []      | Formatted: Line spacing: 1.5 lines       |
|           | Combined                    | 0.904(0.911) | 27.5(31.0)    | 0.90(0.87) | 23.1(27.4) | 34.6%    | 19 ┥     |                     | Formatted: Line spacing: 1.5 lines       |
| Fortress  | 2013/2014                   | 0.881        | 32.4          | 0.92       | 48.0       | -4.5%    | 8 🔹      |                     | Formatted: Line spacing: 1.5 lines       |
| Mountain  | 2012/2014                   | 0.764        | 9 лл   | 0.04       | 56.0       | 2.6%     | 5 4      |                     | Formatted: Line spacing: single          |
| wouldin   | (accumulation)              | 0.704        | 04.4          | 0.94       | 50.0       | -3.0%    | 5        | ), ', ', ( [ | Formatted: Line spacing: 1.5 lines       |
|           | , , , ,                     |              |               |            |            |          |          |                     | Formatted: Line spacing: 1.5 lines       |
|           |                             |              |               |            |            |          |          | N'N C               |                                          |

- 1 Table 2: Regression coefficients and other statistical measures for the multi-season intercomparison of the
- 2 SSG1000 with manual SWE at Sodankylä and Weissfluhjoch (where  $\beta$  and  $\epsilon$  are the slope and intercept of the
- 3 regression line).

|               | Season    | β ε
mm w.e. |       | _2      | DMCC             | Mean   |    |
|---------------|-----------|----------------|-------|---------|------------------|--------|----|
| Site          |           |                | r     | mm w.e. | Relative
Bias | n      |    |
| Sodankylä     | 2013/2014 | 1.05           | -15.5 | 0.84    | 24.2             | -15.1% | 17 |
|               | 2014/2015 | 0.92           | 5.5   | 0.99    | 7.9              | -2.3%  | 10 |
|               | Combined  | 0.99           | -7.3  | 0.88    | 19.8             | -10.8% | 27 |
| Weissfluhjoch | 2013/2014 | 0.72           | 91.7  | 0.97    | 55.5             | 4.2    | 14 |
|               | 2014/2015 | 0.82           | 79.0  | 0.97    | 58.6             | 11.3   | 17 |
|               | Combined  | 0.79           | 77.2  | 0.96    | 57.2             | 8.1    | 31 |

|     | Formatted: Line spacing:    | 1.5 lines |
|-----|-----------------------------|-----------|
|     | Formatted: Left, Line space | ;ing: 1.5 |
|     |                             |           |
| +   | Formatted: Line spacing:    | 1.5 lines |
| (   | ~                           |           |
|     | Formatted: Line spacing:    | 1.5 lines |
| . ( |                             |           |
| 1   | Formatted: Line spacing:    | 1.5 lines |
| (   | ~                           |           |
| 1   | Formatted: Line spacing:    | 1.5 lines |
| (   |                             |           |
|     | Formatted: Line spacing:    | 1.5 lines |
| . ( |                             |           |
|     | Formatted: Line spacing:    | 1.5 lines |

9

5 Table 3: Regression coefficients and coefficient of determination for the multi-season intercomparison of the

6 CS725 with the SSG1000 SWE measurements at Sodankylä (where  $\beta$  and  $\epsilon$  are the slope and intercept of the

7 regression line). Accumulation indicates that data occurring after maximum seasonal SWE is omitted from the

8 analysis.

| Season                      | β    | 3       | r²   |
|-----------------------------|------|---------|------|
|                             | -    | mm w.e. |      |
| 2013/2014                   | 1.20 | 15.7    | 0.90 |
| 2013/2014
(accumulation) | 1.24 | 4.29    | 0.98 |
| 2014/2015                   | 1.19 | 11.9    | 0.99 |

|     | Formatted: Left, Line spacing: 1.5 lines |
|-----|------------------------------------------|
| . Ì | Formatted: Line spacing: 1.5 lines       |
|     | Formatted: Line spacing: 1.5 lines       |
|     | Formatted: Line spacing: single          |
|     | Formatted: Line spacing: 1.5 lines       |
|     | Formatted: Line spacing: 1.5 lines       |
| Ì   | Formatted Table                          |

**1 Figures**

---

## Author Response (AR3)

**An assessment of two automated snow water equivalent instruments during the WMO Solid Precipitation Intercomparison Experiment**

Craig D. Smith, Anna Kontu, Richard Laffin, and John W. Pomeroy

**Response to reviewers #3 comments**

The final comments from reviewer #3 are contained in an annotated manuscript. The comments below reproduce those comments where appropriate and a response is provided.

Page 4, Line 2: I think you could also cite more recent work by Jerry Johnson, e.g.:
Johnson, J. B., Gelvin, A. B., Duvoy, P., Schaefer, G. L., Poole, G. and Horton, G. D.: Performance characteristics of a new electronic snow water equivalent sensor in different climates, Hydrol. Process., 29(6), 1418–1433, doi:10.1002/hyp.10211, 2015.

Response: after reviewing this suggested paper, the reference was added

Page 6, Line 9: If I understand this correctly, the FOV is 120°, following more conventional definitions ...(I don't care about the manufacturer's definition of the coverage angle!)

Response: Changed the reference to the "FOV" to 120°, removed "from centre" and changed the following reference of "FOV" to "response area". This was modified throughout, as suggested and where appropriate.

Page 6, Line 21: I recommend to use consistently 'snowpack'

Response: changed "snow pack" to "snowpack" throughout.

Page 8, Line 16: What about 'real' conditions? Maybe the following paper could be cited here: Sturm, M., Taras, B., Liston, G. E., Derksen, C., Jonas, T. and Lea, J.: Estimating Snow Water Equivalent Using Snow Depth Data and Climate Classes, J. Hydrometeor, 11(6), 1380–1394, doi:10.1175/2010JHM1202.1, 2010.

Response: The statement "…in ideal conditions" was to indicate that the ESC-30, in itself, is considered to be an accurate instrument. A discussion of "real conditions" is included in Section 4.1.5. Added the following statement here: "Errors associated with sampling in less than ideal conditions are discussed later." The suggested reference validates some of the estimates presented in Section 4.1.5 and will be cited in this section.

Page 13, Line 2: Sorry, I guess I missed that one in my first review. Indeed, I expect melting periods to be characterised by both strong enough winds and temperatures above freezing. This may explain why some/many of your 'melting periods' do not match with jumps in Δswe. You should be able to do such an analysis quite quickly, but I don't know if in time for that paper ...

Response: I'm not exactly sure what the reviewer is indicating here, but the purpose of noting the melting temperatures and high wind is to indicate the potential for the formation of ice layers on top and within the snowpack, possibly infiltration of melt water into the frozen sandy soil, both of which could lead to errors discussed in Section 4. I will clarify this.

Page 13, Line 18: I still have hard times to understand how air temperatures raise temperatures within the soil while the latter is still snow covered! Was the snow cover patchy at that time? Did solar radiation penetrate through a shallow snowpack? How do you explain it?

Response: The air temperature doesn't have to raise the temperature of the soil. Infiltration can occur into these frozen soils because of the high sand content. However, the soil temperature sensors indicate that the soil temperatures near the surface rise above 0 C in mid-March. This could be related to patchiness as the reviewer suggests, or an increase in the near surface temperature as a result of melt water. Photos suggest that it could be either.

[Figure]

This is clarified in the manuscript with the following addition: "The near surface (0 to 5 cm) soil temperatures rose above freezing even with snow on the surface. The snowpack was patchy (verified from hourly photos) and shallow, and meltwater was likely percolating through the snow and into the top layers of the soil."

Page 14, Line 2: In your answer you showed that you can estimate GWC from WFV, and vice versa. Why don't you do it here to simplify the comparison for the reader?

Response: As stated before, the soil moisture instrument reports volumetric water content (VWC; note that we updated this from water fraction by volume) while the calibration for the CS725 is entered as gravimetric water content (GWC). They are related through the specific gravity of the soil, which is about 1.4 for loose sand, verified by measurement at a nearby site. Therefore, WFV~GWC*1.4. This is added to the text, including references and the discussion is updated accordingly. After doing the conversion, we can now more accurately state the errors related to change in soil moisture. These errors are slightly lower than those calculated when assuming that GWC~VWC. For example, the increase in soil moisture prior to freeze up only explains 30% of the error, not 40% as previously calculated. These numbers and the discussion are updated accordingly.

Page 14, Line 20: I guess you also need to consider the different ranges for the three sites. My impression is that measurements at Fortress Mountain are not worse than at Caribou Creek, but quite different from Sodankylä (slope).

Response: We are not exactly sure what the reviewer is asking here.

Page 15, Line 23: Insert second sentence of last paragraph of section 4.2 here.

Response: This doesn't really seem to fit well here. We will consider other options

Page 16, Line 13: Errors may compensate each other.

Response: This is probably correct, added this statement in the paragraph

Page 17, Line 8: Is this the mean of RMB? Or is it only for the combined values?
If this is the case, than corresponding r2 are 0.90 and 0.92

Response: Clarified this in the text. The reported "average bias" did indeed only refer to the combined seasons.

Page 20, Line 2: Again, it would be very useful to the reader if he could easily relate GWC to WFV. If you can't provide this relation, I doubt you can calibrate your sensors anyway.

Response: This was fixed, see note above

Page 20, Line 11: As the manual measurements are not shown in Figure 6, it is really hard to follow your argumentation by trying to compare with Figure 4. It looks like the measurement area was snow free around 1 April 2015. Was the snow cover patchy before that date? This could also explain why the soil thawed.

Response: Sorry, Figure 6 was inserted incorrectly into this version. It has now been fixed with the inclusion of the SWE bias. See the response to "Page 12, Line 18" for clarification on why the soil moisture and temperature were changing at this time.

Page 21, Line 5: I'd rather say that further increases due to new snowfall will not be recorded. Thus this would explain a constant value over time. Decreases, however, may rather be due to processes occurring near the soil and/or the sensor surface. See e.g. Johnson, J. B.: A theory of pressure sensor performance in snow, Hydrol. Process., 18(1), 53–64, doi:10.1002/hyp.1310, 2004.

Response: We agree with the reviewer's statement and the sentence "thus potentially decreasing the weight on the pillow of scale" is changed to "causing an underestimate of measured SWE with further accumulation on the surface." The reference was added to point readers towards the physical explanation of these and other errors.

Page 24, Line 13: Which, in view of my own experience and literature (Sturm, 2010), is absolutely realistic!

Response: Sturm (2010) reports an average error of about 7%. This is quoted and referenced.

Page 24, Line 16: Measurements at Sodankylä are coupled to snow pit profiles (like at Weissfluhjoch) that facilitate taking account of thinner ice layers. Thus maybe not that surprising!

Response: Actually, the snow pit measurements are not done in the SPICE intercomparison field and are therefore not coupled to the snow core measurements near the SWE instruments.

Page 25, Line 7: I concur with the editor that the two first paragraphs are results. Thus I'd definitely put them at the end of section 3, but not in the discussion.

The third paragraph is also misplaced in the discussion, still being a mix of setup and results. What about placing the first sentence under section 2.2 on p7, line 13. The second sentence could then be slightly altered and placed at the end of section 3.2.1 on p15, line 23:

"To avoid problems due to flooding of the electronics box, the manufacturer now provides longer cables allowing a better installment of the box."

Response: Moved Section 4.2 to the end of Section 3 to become Section 3.4. Removed the last paragraph in this section which discussed the fix to the SSG1000 failure issue and augmented the last sentence in section 3.2.1 to read "…data are missing due to a sensor failure caused by water damage to the electronics (an issue later remedied by the manufacturer)."

Page 26, Line 11: gain, do a proper comparison as the average SWE values do not correspond to your ranges of $r^2$

Response: To be consistent with the Discussion, the Conclusion states the MRB and correlations for the combined seasons, and then states that the interseasonal variability in both of these measures were higher at Caribou Creek for the reasons stated.

Page 26, Line 18: Which is also due to another range of measured SWE values!

Response: To be consistent, we now state the correlation and mean bias for Fortress Mountain for the whole season with the caveat that there is only one intercomparison season for this site.

Page 27, Line 9-10: Needs to be reformulated in view of other error producing mechanisms.

Response: Revised the statement in question to: "..likely due to bridging of the snowpack on the weighing plate but we also have to consider errors related to the manual measurements and other processes going on at the snow-soil-sensor interface (as outlined in Johnson, 2004)."

Page 28, Line 4: Relative to what. This is unquantified as are any adjectives like good, fair etc. Quantify or reword the sentence.

Response: Replaced "…relatively good agreement…" with "high correlations (generally higher than 0.90, Caribou Creek being the exception)…"

Page 28, Line 7: suggested changing "potential" to "potential error".

Response: done

Page 28, Line 10: This is pure speculation! At least you would need to show that the measurement makes sense, which is not the case in the paper.

Response: We are confident that the increasing error in the CS725 SWE measurements at the end of the season at both Caribou Creek and Sodankyla are because of infiltration into the sand, and we showed that the increase in soil moisture at the end of the ablation season accounts for much of the error. We agree that there is some speculation here because we only have one season of data at one site to back this up, and will therefore change those last sentences to "Although more verification work is required on the impact of soil moisture change on the CS725 bias, aggregating sub-surface moisture in the SWE estimate could potentially be useful from a hydrological perspective as it ultimately impacts the amount of water available for runoff. Nevertheless, it is recommended to co-locate the CS725 with ancillary measurements of soil moisture, soil temperature, and snow depth to guide the user in interpreting the data."

Page 33, Line 2: I'd suggest you explain in the caption why there are values in parentheses. Putting them below the symbols is ambiguous, particularly for MRB, which should have (%) and then no units added to each value. What is n?

Response:  caption now becomes "Table 1: Regression coefficients and other statistical measures for the multi-season intercomparison of the CS725 with manual SWE at Sodankylä, Caribou Creek and Fortress Mountain (where β and ε are the slope and intercept of the regression line). Values inside and outside of the parenthesis represent the respective Thallium and Potassium output from the sensor. "Accumulation" indicates that data occurring after maximum seasonal SWE is omitted from the analysis, "Combined" indicates that data from both seasons are included, and n represents the sample size."  Added "%" to the MRB column heading and removed "%" from the column data.

Page 34, Line 1: What is n? Add (%) to MRB in the top row and delete it from table values.

Response: fixed, as above

Page 35, Line 1: I would get rid of the x-axis legend "Date (UTC)" as it is evident you show a date, However, I'd really prefer having an ISO date format as dd-mm-yyyy instead of the ambiguous mm/dd/yyyy.

Response:  Removed the x-axis title from all time series figures and changed the date format to the ISO 8601 standard of yyyy-mm-dd

Page 38, Figure 5: Y-axis labeling could still be improved by using both axes: left for "Air temperature at 1.5 m (°C)", right for "Difference in SWE (mm w.e.) Also, a better  placement could reduce overlap of panels.

Response:  added a right y-axis for "Instrument Bias" with air temperature remaining on the left y-axis.  Panels no longer overlap.

Page 39, Figure 6: I guess in blue? Where are they?

Response: fixed the caption to say that the soil moisture line is blue.  Oops…the figure in this version was an older figure that didn't have the instrument bias plotted on it…this was fixed.

Page 40, Figure 7: You could use different colors to differentiate between both seasons.

Response: We opted not to do this in previous figures as it made them too busy and difficult to read.  Rather, we have the reader see the seasonal differences in the time series plots following the scatterplot. To be consistent, we combined the two seasons in this plot as well.

Page 42, Figure 9 caption: should be blue dots / blue triangles

Response: fixed

[revised manuscript text omitted]